# E2F1 proteolysis via SCF-cyclin F underlies synthetic lethality between cyclin F loss and Chk1 inhibition

Kamila Burdova[1,†], Hongbin Yang[1,†], Roberta Faedda[1], Samuel Hume[1], Jagat Chauhan[2], Daniel Ebner[3], Benedikt M Kessler[3], Iolanda Vendrell[1,3], David H Drewry[4], Carrow I Wells[4], Stephanie B Hatch[3], Grigory L Dianov[1,5,6], Francesca M Buffa[1] & Vincenzo D'Angiolella[1,*]

## Abstract

Cyclins are central engines of cell cycle progression in conjunction with cyclin-dependent kinases (CDKs). Among the different cyclins controlling cell cycle progression, cyclin F does not partner with a CDK, but instead forms via its F-box domain an SCF (Skp1-Cul1-F-box)-type E3 ubiquitin ligase module. Although various substrates of cyclin F have been identified, the vulnerabilities of cells lacking cyclin F are not known. Thus, we assessed viability of cells lacking cyclin F upon challenging them with more than 180 different kinase inhibitors. The screen revealed a striking synthetic lethality between Chk1 inhibition and cyclin F loss. Chk1 inhibition in cells lacking cyclin F leads to DNA replication catastrophe. Replication catastrophe depends on accumulation of the transcription factor E2F1 in cyclin F-depleted cells. We find that SCF-cyclin F controls E2F1 ubiquitylation and degradation during the G2/M phase of the cell cycle and upon challenging cells with Chk1 inhibitors. Thus, Cyclin F restricts E2F1 activity during the cell cycle and upon checkpoint inhibition to prevent DNA replication stress. Our findings pave the way for patient selection in the clinical use of checkpoint inhibitors.

**Keywords** cell cycle; checkpoints; Chk1; cyclin F; F-box proteins
**Subject Categories** Cell Cycle; DNA Replication & Repair; Post-translational Modifications & Proteolysis
**The EMBO Journal (2019) 38: e101443**

See also: **HI Rösner & CS Sørensen** (October 2019)

## Introduction

Cell cycle transitions are operated by the periodic oscillations of cyclins, which bind cyclin-dependent kinases (CDKs) to promote phosphorylation of target substrates and drive cell cycle progression. Entry into the S phase of the cell cycle is initiated by the activation of the transcription factor E2F1 that promotes accumulation of S phase cyclins and induces the transcription of genes required for DNA replication. The activity of E2F1 is restrained principally by the retinoblastoma (Rb) protein, which masks the E2F1 transactivation domain to keep it inactive. The control of Rb is crucial to prevent unscheduled cell cycle entry, a hallmark of cancer. Indeed, the loss of Rb is a common cancer event (Dyson, 2016).

Among the cyclins co-ordinating cell cycle progression, cyclin E/A and cyclin B, in partnership with CDK2 and CDK1, respectively, promote the progression of cells through S phase and G2 phase (Lim & Kaldis, 2013). Cyclin F is most similar to cyclin A but does not act as an activator of CDKs and is not able to bind to CDKs (Bai *et al*, 1994). Instead, cyclin F, also known as F-box only protein 1 (Fbxo1), is the founding member of the F-box family of proteins (Bai *et al*, 1996). Cyclin F, through the F-box domain, forms a functional Skp1-Cul1-F-Box (SCF) complex acting as an E3 ubiquitin ligase. The SCF[cyclin F] mediates the ubiquitylation and degradation of proteins important for cell cycle progression and genome stability (D'Angiolella *et al*, 2013; Klein *et al*, 2015).

In addition to the coordinated action of cyclins, cell cycle progression is monitored by the presence of checkpoints, which restrict CDK activity and prevent the execution of cell cycle phases if the previous one has not been completed.

Blocks in the progression of DNA replication fork promote the accumulation of single-stranded DNA (ssDNA) behind the fork, whose coating with the ssDNA-binding protein RPA represents a stimulus for the activation of the ATR kinase. Active ATR elicits a

1   Department of Oncology, Medical Research Council Institute for Radiation Oncology, University of Oxford, Oxford, UK
2   Nuffield Department of Clinical Medicine, Ludwig Institute for Cancer Research, University of Oxford, Headington, Oxford, UK
3   Nuffield Department of Medicine, Target Discovery Institute, University of Oxford, Oxford, UK
4   Structural Genomics Consortium, UNC Eshelman School of Pharmacy, University of North Carolina at Chapel Hill, Chapel Hill, NC, USA
5   Institute of Cytology and Genetics, Russian Academy of Sciences, Novosibirsk, Russian Federation
6   Novosibirsk State University, Novosibirsk, Russian Federation
    *Corresponding author. Tel: +44 1865 617400; E-mail: vincenzo.dangiolella@oncology.ox.ac.uk
    †These authors contributed equally to this work

checkpoint response by phosphorylating Chk1 and initiating the signalling cascade that culminates with CDK inactivation. Chk1 exerts this function by controlling the ubiquitylation and subsequent degradation of Cdc25A phosphatase (Busino *et al*, 2003; Jin *et al*, 2003; Kotsantis *et al*, 2018). Cdc25A removes the inhibitory phosphorylation on Tyr15 of CDKs, mediated by the tyrosine kinase Wee1 to prompt CDK activation. Upon activation of Chk1, the degradation of Cdc25A is initiated and CDK activity is suppressed to restrain DNA replication and progression of the cell cycle (Bartek & Lukas, 2003). The checkpoint response allows DNA repair before committing to mitosis by restricting CDK activity until the S/G2 transition through Chk1 (Lemmens *et al*, 2018). If the blocks at the replication fork cannot be bypassed or repaired, the ssDNA gaps can be converted into double-stranded breaks (DSBs), which are more deleterious and could lead to gross chromosomal rearrangements resulting in cell death. Given their high proliferative capacity and compromised DNA repair, cancer cells have intrinsically higher replication stress and DNA damage. These observations have spurred the development of a number of inhibitors targeting the checkpoint response mediated by ATR and Chk1. Although ATR and Chk1 inhibitors are being evaluated in multiple clinical trials, the genetic predispositions that would sensitise or desensitise cancer cells to ATR and Chk1 inhibitors are not fully elucidated.

By screening the kinase chemogenomic set (KCGS) to identify vulnerabilities of cells lacking cyclin F, we discover a novel synthetic lethal interaction between cyclin F loss and Chk1 inhibition. We observe that the loss of cyclin F promotes DNA replication catastrophe after challenging cells with Chk1 inhibitors. The DNA replication catastrophe in cells lacking cyclin F after Chk1 inhibitors is promoted by the accumulation of E2F1. E2F1 is a novel substrate of cyclin F which is controlled at the G2/M transition and after checkpoint inhibition. By expressing a non-degradable version of E2F1, we show that E2F1 accumulation exacerbates Chk1 inhibitor sensitivity, promoting the accumulation of DSBs and cell death.

## Results

### KCGS screen identifies synthetic lethality between Chk1 inhibitors and cyclin F loss

Using CRISPR (clustered regularly interspaced short palindromic repeats)/Cas9, we have previously generated cyclin F knock-out (*CCNF* K/O) cell lines (Mavrommati *et al*, 2018). Interestingly, these cells present a cell cycle profile comparable to the control parental cell line (Fig EV1A and B). To identify weaknesses of cells lacking cyclin F, we conducted a kinase inhibitor drug screen to assess differences in viability between *CCNF* K/O and parental cell lines (HeLa; Fig 1A). To this end, we used the KCGS, a set whose origins can be traced to the well-utilised kinase inhibitor collections PKIS and PKIS2 (Elkins *et al*, 2016; Drewry *et al*, 2017). The simple premise of KCGS is that screening a publicly available, well annotated set of potent and selective kinase inhibitors in disease-relevant phenotypic assays is an efficient way to elucidate biology and uncover dependencies (Jones & Bunnage, 2017). The current version of KCGS consists of 188 small molecule kinase inhibitors that cover 221 kinases. None of the members of KCGS is broadly

kinome active, facilitating target deconvolution. The screen was run in duplicate with an average replicate correlation (Pearsons') of 0.9467 and an average Z-Factor of 0.6040, demonstrating a good dynamic range between DMSO (vehicle-negative control) and controls (chemo-toxic-positive control) with robust reproducibility between replicates. To identify small molecule kinase inhibitors, which specifically target cyclin F-depleted cells, we normalised the cell viability results using a Z-score for each cell line and then expressed the difference between *CCNF* K/O cells and control cells as $\Delta Z$. The graph in Fig 1A and additional data in Table 1 highlight compounds to which *CCNF* K/O cells are sensitive or resistant (Dataset EV1). The most striking difference in viability between control cells and *CCNF* K/O cells was obtained with CCT244747, a selective Chk1 kinase inhibitor (Walton *et al*, 2012). Among the compounds which killed *CCNF* K/O cells, we also identified VE-822, an ATR inhibitor (Charrier *et al*, 2011). Similarly, in a parallel drug screen, a striking sensitivity of *CCNF* K/O cells to two other Chk1 inhibitors with unrelated structure, PF477736 and AZD7762, was observed (Fig 1B; Blasina *et al*, 2008; Zabludoff *et al*, 2008).

To confirm that the effect of CCT244747, PF47736 and AZD7762 was on target, we also tested the survival of the HeLa *CCNF* K/O with LY2603618 and UCN-01, two structurally unrelated Chk1 inhibitors. UCN-01 decreased cell proliferation in *CCNF* K/O cells compared to HeLa parental cells (Fig 1C). Upon treatment of cells with LY2603618, a very selective Chk1 inhibitor (King *et al*, 2014), we observed the most striking difference in cell viability. The IC50 of LY2603618 in normal HeLa cells was > 1 μM, whereas *CCNF* K/O cells had an IC50 of 400 nM, accounting for a significant fold difference in sensitivity across the two cell lines using LY2603618 (Fig 1D).

To establish that the difference in sensitivity between control and *CCNF* K/O cells was not due to proliferation defects of the *CCNF* K/O cells, we measured cell proliferation across 72 h comparing HeLa control to *CCNF* K/O. Overall, cell proliferation was not different in the 72-h time-frame between HeLa control and *CCNF* K/O (Fig 1E). However, upon addition of LY2603618, *CCNF* K/O cells stopped proliferating (Fig 1E), corresponding to an increase in cell death (Fig 1F).

To exclude the possibility that the phenotype of cell death induced by Chk1 inhibitors was limited to *CCNF* K/O cell lines, we also tested sensitivity of U-2-OS cells to LY2603618 after knockdown of cyclin F by siRNA. Although treatment of U-2-OS cells with LY2603618 at 500 nM caused more cell death in U-2-OS cells compared to HeLa cells, U-2-OS cells where cyclin F levels were diminished by siRNA had significantly reduced cell survival and increased cell death (Fig 1G and H). In addition, non-transformed cell line RPE-1 showed increased sensitivity to Chk1 inhibitors upon siRNA of cyclin F (Fig EV1C).

Since ATR is upstream of Chk1, it was not surprising that the drug screen above also identified ATR inhibitors. Indeed, inhibition of ATR using VE-821 also promoted cell killing in the *CCNF* K/O, although the difference between HeLa control and *CCNF* K/O cells was only observed at high doses and was less pronounced compared to Chk1 inhibitors (Fig EV1D). Similar results were obtained after inhibition of ATR in U-2-OS cells and cyclin F depletion by siRNA (Fig EV1E). The data presented above highlight a novel synthetic lethal interaction between the loss of cyclin F and checkpoint depletion. The sensitivity of cyclin F loss to Chk1 inhibition was striking and investigated further.

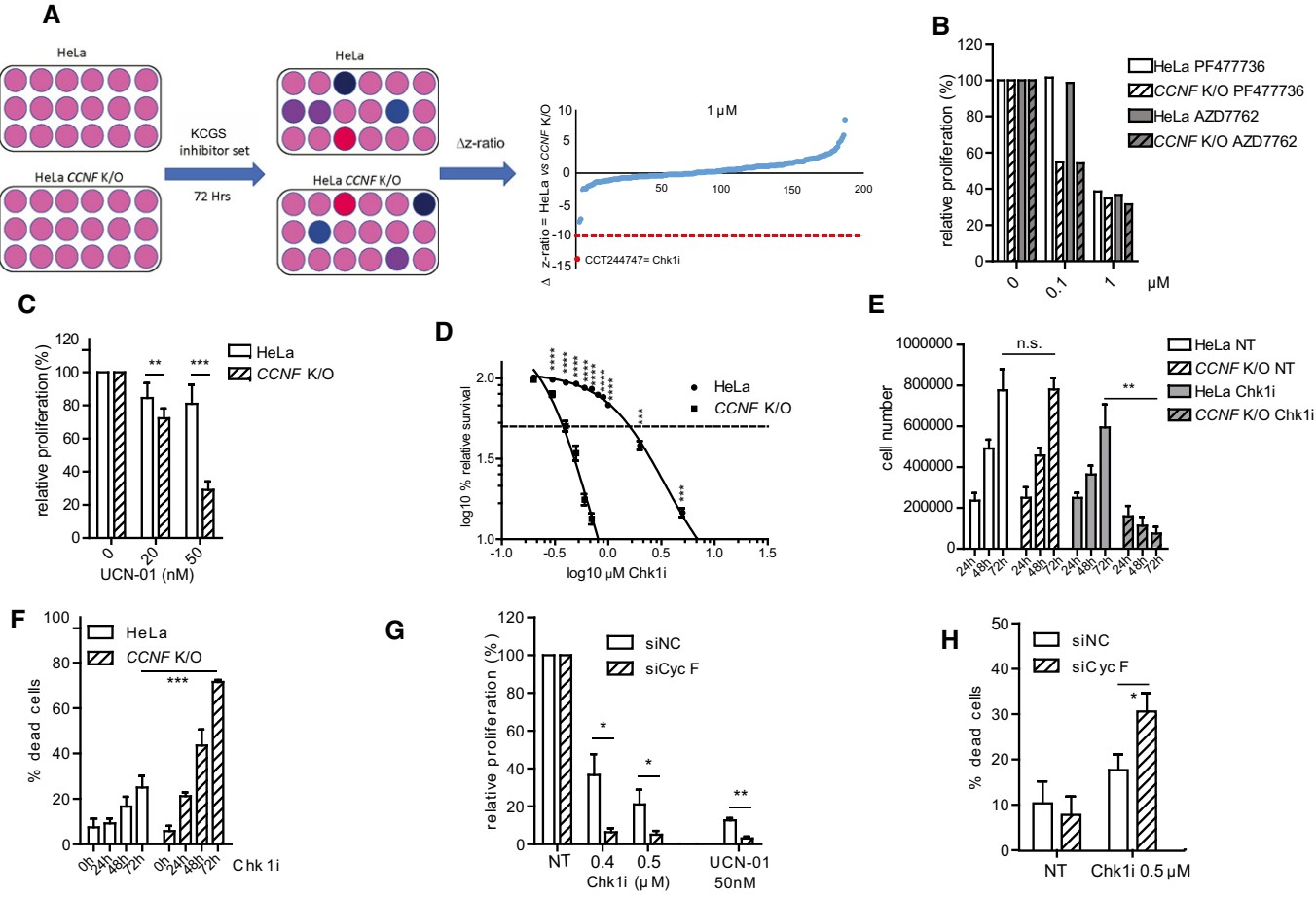

**Figure 1. KCGS screen identifies synthetic lethality between Chk1 inhibitors and cyclin F loss.**

A HeLa and cyclin F knock-out (*CCNF* K/O) cells were treated with the kinase chemogenomic set (KSGS) in 384-well format. After 72 h, viability was measured using resazurin. Flowchart representation (*left*). The robust *Z*-score difference is plotted for data acquired at 1 μM concentration (*right*).

B Cell survival measured using resazurin and compared to controls treated with DMSO (expressed as relative proliferation %). Cells were treated with the indicated inhibitors at 0.1 and 1 μM concentration.

C Cell survival measured using resazurin and compared to controls (0) treated with DMSO (expressed as relative proliferation %). Cells were treated with UCN-01 at the indicated concentrations.

D Differences in viability of HeLa and *CCNF* K/O cells after treatment with specific Chk1i (LY2603618) at the indicated concentrations plotted on a log$_{10}$ scale to measure differences in IC50.

E Number of HeLa and *CCNF* K/O cells left untreated or treated with Chk1i (LY2603618) for 24, 48 or 72 h as indicated. (n.s. = non-significant).

F Cell viability measurements (expressed as number of dead cells %) using propidium iodide staining for HeLa and *CCNF* K/O cells after treatment with Chk1i (LY2603618).

G U-2-OS cells were seeded and transfected with non-targeting siRNA siNC (negative control) or siCyc F. Twenty-four hours after transfection, cells were treated with UCN-01 or Chk1i (LY2603618) at indicated concentrations for 3 days before viability was measured.

H Cell viability measurements LY2603618 using propidium iodide staining in U-2-OS cells transfected with non-targeting siNC and siCyc F and treated with Chk1i (LY2603618) as indicated.

Data information: Data are presented as mean ± standard deviation (SD), with at least three independent experiments. *P*-values (**P* < 0.05, ***P* < 0.005, ****P* < 0.0005, *****P* < 0.00005) were calculated by paired and two-tailed *t*-test. NT = treatment with DMSO.

## Loss of cyclin F causes DNA replication catastrophe after Chk1 inhibition

The inhibition of ATR and Chk1 promotes accumulation of ssDNA, which is exacerbated over time due to active DNA replication. The ssDNA saturates the capacity of RPA to bind to it and is exposed to the action of nucleases. Nucleases convert ssDNA to DSBs, which promote the activation of DNA-PK and ATM kinases required for DSB repair. The accumulation of DSBs can be detected by markers

of DNA-PK and ATM kinase activation and is an indication of DNA replication catastrophe (Toledo *et al*, 2013).

To gain insights into the mechanism of synthetic lethality between loss of cyclin F and Chk1 inhibition, we compared the response of HeLa parental cells and *CCNF* K/O cells after treatment with Chk1 inhibitor. In contrast to parental cells that showed accumulation of RPA phosphorylation on Serine (S) 33, a surrogate marker of ATR activation, at 6 h, *CCNF* K/O cells already showed increased RPA S33 phosphorylation after 2 h of treatment (Fig 2A).

**Table 1.  Top hits of the viability screen comparing HeLa to *CCNF* K/O using the KCGS.**

| | Compound name | Δ z-score | Target |
|---|---|---|---|
| **100 nM** | | | |
| **More sensitive** | | | |
| C10 | VE-822 | −0.61484 | ATR |
| B17 | CCT251545 | −0.56548 | CDK8, CDK19 |
| I03 | GSK2336394 | −0.4252 | PIK3CB |
| **Less sensitive** | | | |
| A20 | GW784752X | 1.016439 | GSK3b |
| A21 | GW580509X | 1.052113 | VEGFR2 |
| E22 | GW296115X | 1.443446 | PDGFR |
| **1 µM** | | | |
| **More sensitive** | | | |
| B08 | CCT244747 | −2.88143 | CHK1 |
| E16 | BI00614644 | −0.56505 | MAPKAPK2, MAPKAPK5, PIM3, PHKG2 |
| I05 | SB-725317 | −0.55738 | CDK2/cyclinA, LTK |
| **Less sensitive** | | | |
| C11 | SB-590885 | 1.036447 | B-raf |
| B03 | JNK-IN-7 | 1.164645 | JNK |
| E22 | GW296115X | 1.180556 | PDGFRB |
| D15 | GW814408X | 1.364238 | GSK3b |
| H16 | GSK1070916 | 1.572064 | Aurora kinase |
| D12 | CA93.0 | 2.072805 | GAK1 |
| **10 µM** | | | |
| **More sensitive** | | | |
| H19 | GW807982X | −1.06323 | GSK3b |
| I20 | TPKI-39 | −0.86181 | CIT, KIT, PDGFRB, CSF1R, PDGFRA, DDR1, FLT1, TIE2 |
| C13 | GSK1838705 | −0.73301 | IGF1R, IR |
| **Less sensitive** | | | |
| G14 | PFE-PKIS 3 | 0.838916 | TNK2, TYRO3, YES, FGFR1,2,&3 |
| H11 | CCT241533 | 0.859239 | CHK2 |
| G09 | GW440135 | 0.866643 | KIT, NEK7, MEK5, PDGFRB, VEGFR2, NEK9, PDGFRA, FLT4 |

While ATR activation can be attributed to accumulation of RPA-coated ssDNA, further activation of DNA-PK and ATM can be associated with formation of DSBs leading to DNA replication catastrophe detected by phosphorylation of γ-H2AX, phosphorylation of RPA on S4 and S8 and phosphorylation of KAP1 on S824 at the 16-h time point as reported previously (Toledo *et al*, 2013). In the *CCNF* K/O cell line, phosphorylation of γ-H2AX, phosphorylation of RPA on S4 and S8 and phosphorylation of KAP1 on S824 were already present at 6 h after treatment with the Chk1 inhibitor LY2603618. Activation of the DSB markers continued thereafter, accumulating

profoundly at 16 and 24 h of treatment compared to control cells (Fig 2A). It is worth mentioning that the overall levels of Chk1 and RPA were unchanged in *CCNF* K/O cells compared to control cells, indicating that the differences observed were due to upstream phosphorylation events rather than a change in the total levels of these proteins (Fig 2A). The phenotype of significant accumulation of single-stranded DNA (ssDNA) and DSB markers was reproduced in the U-2-OS cell line by knockdown of cyclin F using siRNA in the presence of Chk1 inhibitors (Fig EV2A).

To confirm that *CCNF* K/O cells indeed encountered DNA replication catastrophe, we analysed chromatin bound RPA and γ-H2AX formation in response to Chk1 inhibition in different cell cycle phases by Fluorescence Activated Cell Sorting (FACS). In accordance with the induction of profound DNA replication stress in the *CCNF* K/O, the γ-H2AX signal initiated in replicating cells (Fig 2B–D) at early time points, and γ-H2AX presence and levels increased as cells progressed from early S to mid-late S phase (Figs 2E and F, and EV2B and C). At 24 h after Chk1 inhibitor treatment, about 15% of cells were found to be highly γ-H2AX positive but EdU negative, having S phase DNA content of 2-4n (Fig 2G). These cells are arrested in S phase, likely due to the extensive accumulation of DNA damage detected by γ-H2AX staining. In addition, we observed excessive RPA loading on chromatin starting from 6 h of treatment of Chk1 inhibitors in *CCNF* K/O cells (Fig EV2D). Consistent with the previous observation of DNA replication catastrophe induced by checkpoint inhibition, the accumulation of RPA on chromatin reached its peak before DSB generation (Fig EV2D). The conversion of ssDNA (high RPA loading) to DSBs (γ-H2AX) was confined to cells with the highest degree of chromatin-loaded RPA (Fig 2H). The shape of the plot of the nuclear ratio of RPA to γ-H2AX following Chk1 inhibition in *CCNF* K/O treatment was consistent with DNA replication catastrophe induced by checkpoint inhibition (Fig 2H).

It is important to mention that γ-H2AX formation in *CCNF* K/O cells upon Chk1 inhibitor treatment was significantly reduced after expression of GFP-cyclin F (Fig 3A and B), demonstrating that the formation of DSBs is mediated specifically by cyclin F loss.

Although the evidence provided so far is indicative of DNA replication catastrophe in *CCNF* K/O cells upon Chk1 inhibition, we sought direct evidence of single-stranded breaks (SSBs) and DSB accumulation. Therefore, we conducted alkaline and neutral comet assays in *CCNF* K/O cells treated with Chk1 inhibitors. Treatment of *CCNF* K/O cells with Chk1 inhibitors induced a drastic increase in SSBs and DSBs readily visible in the tail images (Fig 3C) and quantified in the olive tail moment (Fig 3D and E). Our data support a model where loss of cyclin F predisposes cells to DNA replication catastrophe upon challenging cells with Chk1 inhibitors.

### E2F1 promotes DNA replication catastrophe in *CCNF* K/O cells treated with Chk1 inhibitors

To identify novel substrates of cyclin F that mediate DNA replication stress after checkpoint inhibition, we performed immunoprecipitation of cyclin F and identified interacting partners by liquid chromatography-tandem mass spectrometry (LC-MS/MS). The LC-MS/MS was conducted in the absence and presence of MLN4924, an inhibitor of Nedd8 activating enzyme, which prevents the activity of SCF complexes (Soucy *et al*, 2009). Results are summarised in Table 2 and include components of G1/S phase transition regulators and

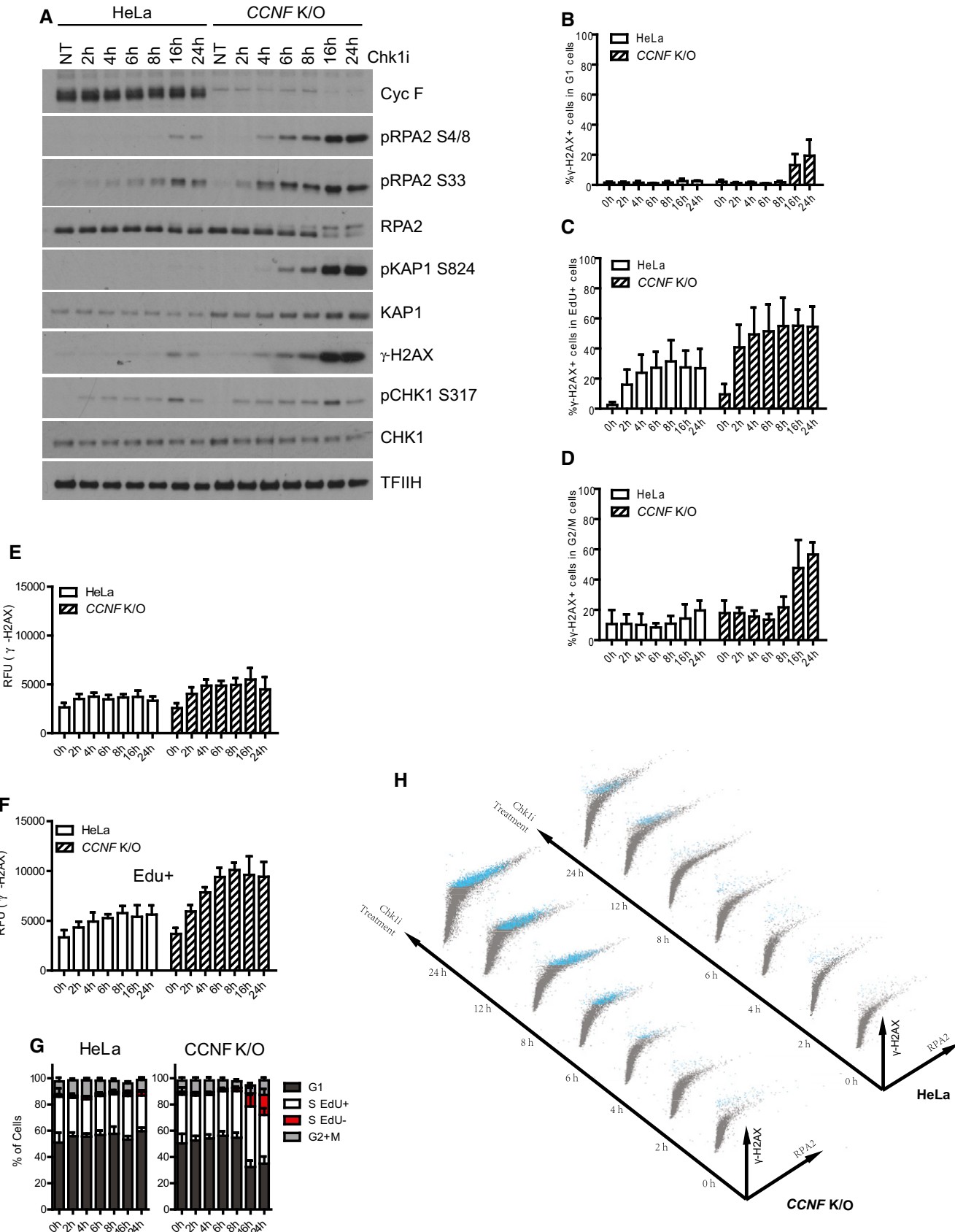

**Figure 2.**

**Figure 2. Loss of cyclin F causes DNA replication catastrophe after Chk1 inhibition.**

A  Cells treated with 1 μM Chk1i (LY2603618) for the indicated hours (h) were harvested and lysed using SDS. Indicated proteins were resolved by SDS–PAGE and detected by Western blot (WB). TFIIH was used as a loading control.

B  Percentage (%) of γ-H2AX-positive cells in G1 cells after Chk1i (LY2603618) treatment for the indicated hours (h) measured using Fluorescence Activated Cell Sorting (FACS). G1 cells were considered having DAPI 2n staining and EdU negative.

C  Percentage (%) of γ-H2AX-positive cells in replicating (EdU⁺) cells after Chk1i (LY2603618) treatment at the indicated time points.

D  Percentage (%) of γ-H2AX-positive cells in G2/M cells after Chk1i (LY2603618) treatment. G2/M cells were considered having DAPI 4n staining and EdU negativity.

E  Intensity of γ-H2AX measured as relative fluorescence units (RFU) in early S phase cells treated with Chk1i. Mean intensity of γ-H2AX fluorescence was evaluated from EdU⁺ cells with 2N DAPI content.

F  Intensity of γ-H2AX measured as relative fluorescence units (RFU) in replicating cells treated with Chk1i. Mean intensity of γ-H2AX fluorescence was evaluated from EdU⁺.

G  Overall cell cycle profile of HeLa and *CCNF* K/O cells after Chk1i treatment. Cells with 2n DAPI staining but EdU⁻ were gated as G1 cells; cells with > 2n but < 4n DAPI staining but EdU⁻ were gated as EdU⁻ S cells; cells with > 2n but < 4n DAPI staining but EdU⁺ were gated as EdU⁺ S phase; cells with 4n DAPI and pH3 S10-negative staining were gated as G2 phase and cells with 4n DAPI and pH3 S10 positive as M cells. Data from at least three independent replicate experiments were plotted as Mean ± SD.

H  Representative plots of γ-H2AX versus RPA2 signal of HeLa and *CCNF* K/O cells after treatment with 1 μM Chk1i (LY2603618) at the indicated hours (h). Blue labelled dots represent γ-H2AX-positive cells. Relates to Materials section for details on representation.

Data information: Data from at least three independent biological replicates were plotted with Mean ± SD (B–D) or Mean RFU ± SD (E, F).

many components of DNA replication. We speculated that these interacting partners accumulate in *CCNF* K/O cell lines and exacerbate DNA replication stress upon Chk1 inhibition. Since components of pre-RC (Cdc6, Cdt1 and Geminin), components of the DNA replication elongation machinery (TICRR and MTBP), G1/S cell cycle transcriptional regulators (E2F1 and E2F3) and exonucleases (Exo1) have all been previously linked to the induction of DNA replication stress, we focused our study on these factors (Hills & Diffley, 2014; Kotsantis *et al*, 2018). It is important to mention that although E2F1 was not identified in the LC-MS/MS, E2F1 deregulation by Rb loss was previously shown to promote synthetic lethality in combination with Chk1 inhibition (Witkiewicz *et al*, 2018). To seek a potential substrate of cyclin F which could mediate sensitivity to Chk1 inhibitors, we performed siRNA of Exo1, Cdc6, Cdt1, Geminin, MTBP, TICRR, E2F1 and E3F3 in HeLa control and *CCNF* K/O cells (Fig EV3A–D) after treatment with Chk1 inhibitors and determined the extent of DNA replication stress by measuring γ-H2AX levels and RPA accumulation on chromatin, respectively, by FACS. We obtained a complete rescue of γ-H2AX and RPA accumulation only after siRNA of E2F1 but not after siRNA of any factor indicated above, suggesting that E2F1 was the main determinant of Chk1 inhibitor sensitivity upon cyclin F depletion (Fig 4A and B). Upon close observation of the DNA damage present exclusively in S phase cells, we noted that siRNA of E2F1 specifically resulted in a reduction of DNA damage in *CCNF* K/O cells (Fig 4C). In addition, E2F1 depletion did not rescue DNA damage in HeLa parental cells (Fig 4C). This experiment points out that the rescue effect of E2F1 depletion on DNA damage is not simply due to cell cycle arrest and is a specific feature of *CCNF* K/O cells. A different siRNA oligo targeting E2F1 but not an siRNA targeting Exo1 rescued sensitivity of *CCNF* K/O cells to Chk1 inhibitors (Fig EV3E). Furthermore, high expression of exogenous E2F1 in HeLa cells was sufficient to mediate DNA replication stress after Chk1 inhibition, as visualised by increased levels of γ-H2AX intensity detected by FACS analysis (Figs 4D and EV3F). All the surrogate markers of ssDNA and DSB formation (phosphorylation of RPA on S33, phosphorylation of RPA on S4 and S8, γ-H2AX and phosphorylation of KAP1 at S824) were increased in *CCNF* K/O cells upon treatment with Chk1 inhibitors as observed above. However, siRNA of E2F1 was sufficient to restore the levels of these markers to basal levels present in HeLa controls (Fig 4E).

A similar set of experiments to the ones described above was conducted upon siRNA of cyclin F and treatment with ATR inhibitors. We observed DNA replication stress upon cyclin F depletion and ATR inhibition, as detected by the appearance of phosphorylation of RPA on S4, S8 and γ-H2AX. The DNA replication stress observed in cyclin F-depleted cells treated with ATR inhibitor was rescued by concomitant depletion of E2F1 (Fig EV3G). Furthermore, we conducted a comet assay to detect directly DNA damage in cells treated as above. The comet assay highlighted that ATR inhibition induced increased DNA damage after siRNA of cyclin F. The DNA damage present in cells with reduced cyclin F levels was rescued fully by concomitant depletion of E2F1 (Fig EV3H). *CCNF* K/O cells died more upon ATR inhibition (Fig EV3I); however, the viability of cells lacking cyclin F upon ATR inhibition was fully rescued by concomitant depletion of E2F1 (Fig EV3I). We conclude that DNA replication catastrophe induced by cyclin F depletion and checkpoint inhibition is promoted by E2F1.

### E2F1 is a ubiquitylation substrate of SCF^cyclin F degraded at the G2/M transition and after checkpoint inhibitors

The experiments presented above suggest that E2F1 could be accumulating in *CCNF* K/O cells. Therefore, we asked whether E2F1 was regulated by cyclin F and measured the levels of E2F1 in *CCNF* K/O cells or after knockdown of cyclin F by siRNA. *CCNF* K/O cell lines had a prominent increase in the levels of E2F1 compared to control (Fig 5A). Furthermore, siRNA of cyclin F induced a significant increase of E2F1 in U-2-OS cells (Fig 5B).

We used normalised gene expression data to compare HeLa cells with the *CCNF* K/O in order to identify differentially expressed genes. Genes with adjusted $P < 0.05$ were considered significantly differentially expressed (Dataset EV2). Gene set variance analysis (GSVA) of three different data sets related to E2F1 and E2Fs target genes (Ishida *et al*, 2001; Schaefer *et al*, 2009; Liberzon *et al*, 2015) was significantly increased in *CCNF* K/O cells compared to HeLa (Fig 5C). Gene set enrichment analysis (GSEA) confirmed a statistically significant enrichment of these gene sets (Fig EV4A). Our data point out that E2F1 protein levels and target genes are increased in *CCNF* K/O cells.

E2F1-dependent transcription is required to promote G1/S phase transition, and the activity of E2F1 is reduced in G2/M

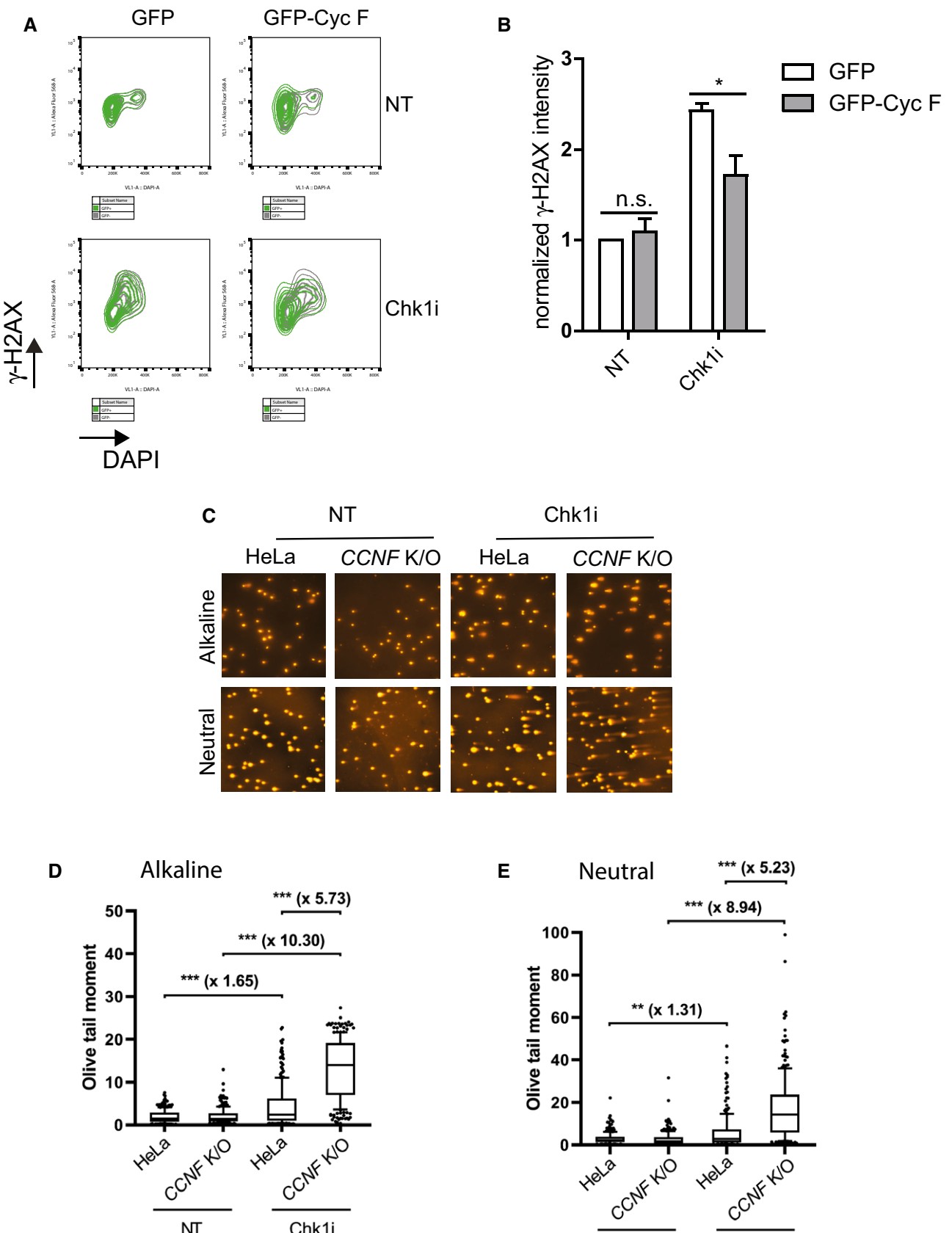

Figure 3.

**Figure 3. Loss of cyclin F causes SSBs and DSBs after Chk1 inhibition.**

A   Representative plots of γ-H2AX intensity versus DAPI signal of *CCNF* K/O cells transfected with empty GFP and GFP-*CCNF* untreated (*top panels*) and after treatment with 1 μM Chk1i (LY2603618) for 20 h (*bottom panels*).

B   Normalised γ-H2AX intensity in *CCNF* K/O cells transfected with either GFP empty vector or GFP-Cyc F and treated with 1 μM Chk1i for 20 h. GFP-positive cells were gated. Data from three independent experiments were plotted with mean % ± SD.

C   Representative images of neutral and alkaline comet assays from HeLa and *CCNF* K/O cells treated with Chk1i. Cells were treated with 1 μM Chk1i for 24 h before being harvested for neutral and alkaline comet assays as indicated.

D   Quantification of comets in *alkaline* gels. At least 100 cells, across two slides, were analysed in each condition in two biological replicates. Data are shown as medians, with 25/75% percentile range (box) and 10–90% percentile range (whiskers). Fold changes of median values are shown in bold. *P*-values were calculated using the Mann–Whitney test (two-tailed). Olive tail moment = (Tail.mean − Head.mean)*Tail%DNA/100.

E   Quantification of comets in *neutral* gels. At least 100 cells, across two slides, were analysed in each condition in two biological replicates. Data are shown as medians, with 25/75% percentile range (box) and 10–90% percentile range (whiskers). Fold changes of median values are shown in bold. *P*-values were calculated using the Mann–Whitney test (two-tailed). Olive tail moment = (Tail.mean − Head.mean)*Tail%DNA/100.

**Table 2. High confidence interacting partners/substrates of cyclin F. Data shown correspond to MASCOT protein outputs including: protein score, N. Significant sequences and emPAI value.**

| Name | Cyclin F | | | | Cyclin F + MLN4924 | | |
|---|---|---|---|---|---|---|---|
| | Score | No.sign peptides | Empai | | Score | No.sign peptides | Empai |
| Cyclin F (F-box only protein 1) | 1,973 | 18 | 2.04 | | 1,993 | 28 | 5.39 |
| Cell division control protein 6 homolog (CDC6-related protein) (Cdc18-related protein) (HsCdc18) [p62(cdc6)] (HsCDC6) | 107 | 3 | 0.25 | | 193 | 9 | 0.95 |
| M-phase phosphoprotein 9 | 56 | 2 | 0.07 | | 162 | 8 | 0.33 |
| Geminin | 52 | 1 | 0.22 | | 247 | 8 | 7.88 |
| Ankyrin repeat domain-containing protein 17 (gene trap ankyrin repeat protein) (serologically defined breast cancer antigen NY-BR-16) | 51 | 1 | 0.02 | | 100 | 5 | 0.09 |
| Zinc finger protein 318 (endocrine regulatory protein) | 50 | 3 | 0.06 | | 126 | 8 | 0.16 |
| Exonuclease 1 (hExo1) (EC 3.1.-.-) (exonuclease I) (hExoI) | 50 | 2 | 0.1 | | 113 | 6 | 0.35 |
| Protein red (cytokine IK) (IK factor) (protein RER) | 45 | 1 | 0.07 | | 53 | 2 | 0.15 |
| DNA replication factor Cdt1 (double-parked homolog) (DUP) | 71 | 2 | 0.17 | | 140 | 10 | 1.17 |
| Opioid growth factor receptor (OGFr) (protein 7-60) (zeta-type opioid receptor) | 60 | 1 | 0.07 | | 37 | 1 | 0.07 |
| Heterogeneous nuclear ribonucleoprotein A3 (hnRNP A3) | 59 | 1 | 0.13 | | 41 | 1 | 0.13 |
| Cullin-1 (CUL-1) | 53 | 2 | 0.11 | | 91 | 3 | 0.17 |
| Ribonucleoside-diphosphate reductase subunit M2 (EC 1.17.4.1) (ribonucleotide reductase small chain) (ribonucleotide reductase small subunit) | 1 | 0 | 0 | | 61 | 1 | 0.11 |
| Treslin (TopBP1-interacting checkpoint and replication regulator) (TopBP1-interacting, replication-stimulating protein) | 31 | 1 | 0.02 | | 68 | 2 | 0.05 |
| Casein kinase II subunit alpha 3 (CK II alpha 3) (EC 2.7.11.1) (casein kinase II alpha 1 polypeptide pseudogene) | 1 | 0 | 0 | | 50 | 1 | 0.12 |
| BRISC and BRCA1-A complex member 1 (mediator of RAP80 interactions and targeting subunit of 40 kDa) (new component of the BRCA1-A complex) | 45 | 1 | 0.14 | | 1 | 0 | 0 |
| Zinc finger protein 8 (zinc finger protein HF.18) | 1 | 0 | 0 | | 39 | 3 | 0.24 |
| Protein aurora borealis (HsBora) | 1 | 0 | 0 | | 38 | 2 | 0.17 |
| Mdm2-binding protein (hMTBP) | 1 | 0 | 0 | | 46 | 1 | 0.05 |
| Transcription factor E2F3 (E2F-3) | 1 | 0 | 0 | | 42 | 1 | 0.1 |
| Transcription factor ETV6 (ETS translocation variant 6) (ETS-related protein Tel1) (Tel) | 1 | 0 | 0 | | 37 | 1 | 0.09 |

(Budhavarapu *et al*, 2012). We speculated that E2F1, like other cyclin F substrates, was degraded at the G2/M transition so we measured levels of E2F1 in synchronised cells. While in HeLa cells the levels of E2F1 were high in S phase and reduced in G2/M, in *CCNF* K/O cells E2F1 levels remained high across the cell cycle (Fig EV4B).

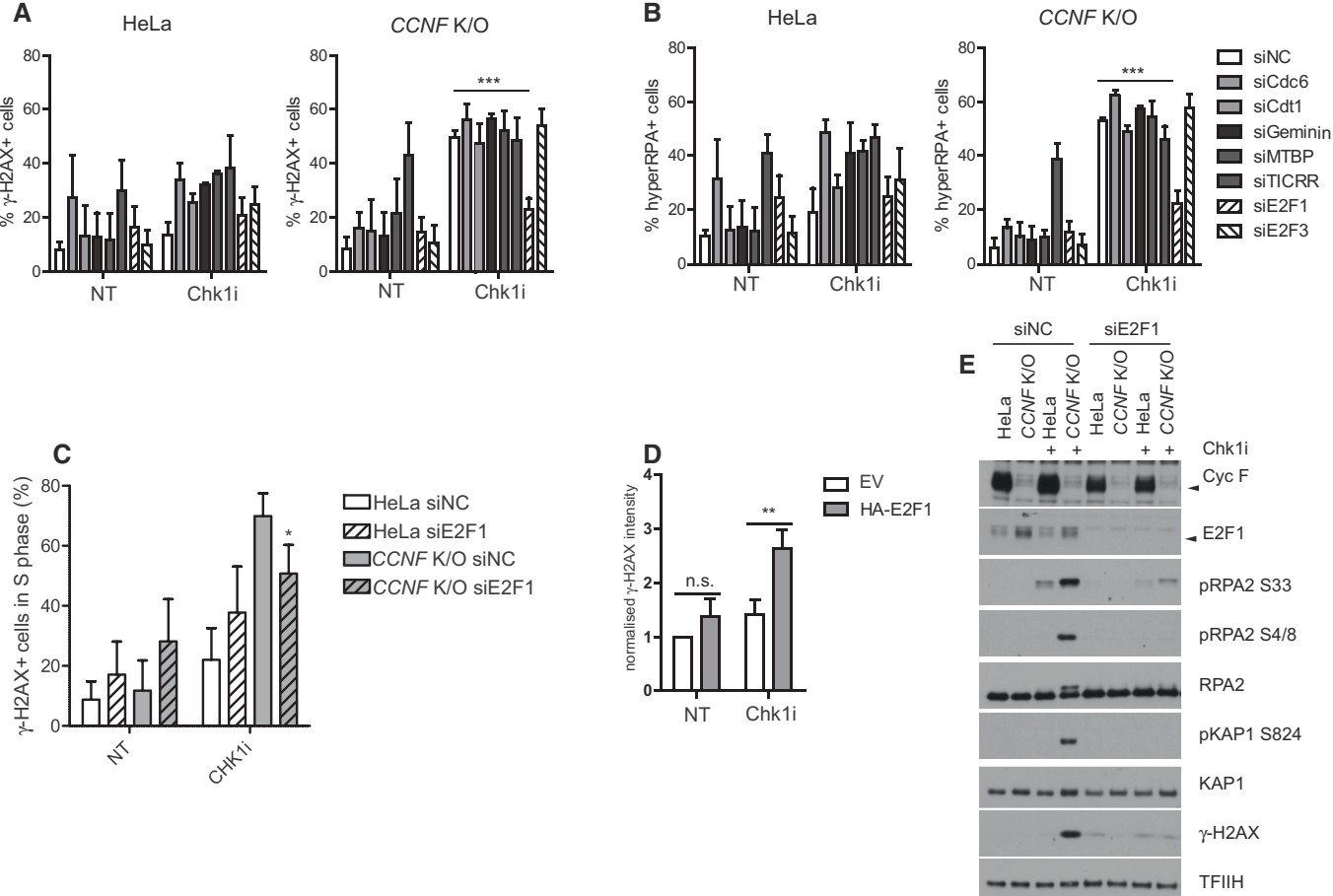

**Figure 4. E2F1 mediates DNA replication catastrophe in cyclin F K/O cells treated with Chk1 inhibitors.**

A   Percentage (%) of γ-H2AX-positive cells in HeLa and *CCNF* K/O cell lines after transfection of the indicated siRNAs and treatment with Chk1i for 20 h. Percent of γ-H2Ax-positive cells was plotted as mean % ± SD.

B   Percentage (%) of hyper-RPA-positive cells in HeLa and *CCNF* K/O cell lines after transfection of the indicated siRNAs and treatment with Chk1i for 20 h. Percent of hyper-RPA-positive cells was plotted as mean % ± SD.

C   Percentage (%) of γ-H2AX-positive cells in S phase cells by DAPI content after Chk1i (LY2603618). Related to the experiments presented in (A).

D   Relative γ-H2AX intensity in cells transfected with either empty vector (EV) or HA-E2F1, treated with Chk1i for 20 h. γ-H2AX fluorescence in HA-positive cells was normalised to the empty vector (EV) transfected cells and compared to cells not expressing HA. Related to Fig EV3F.

E   HeLa and *CCNF* K/O cells transfected with the indicated siRNA and treated with 1 μM Chk1i (LY2603618) for 20 h were harvested and lysed using SDS. Indicated proteins were resolved by SDS–PAGE and detected by WB. TFIIH is a loading control.

Data information: Data are presented as mean ± SD, with at least three independent experiments. *P*-values (**P* < 0.05, ***P* < 0.005, ****P* < 0.0005) were calculated by paired and two-tailed *t*-test.

The experiments presented above show that E2F1 levels and transcriptional targets are increased in *CCNF* K/O cells; therefore, we tested whether E2F1 is directly regulated by cyclin F through ubiquitylation. To prove that the regulation of E2F1 by cyclin F was direct, we tested by immunoprecipitation whether E2F1 could interact with cyclin F. Cyclin F interacts specifically with E2F1, and the interaction between cyclin F and E2F1 was reinforced by treating cells with MLN4924, an inhibitor of Nedd8 activating enzyme, which prevents the activity of SCF complexes (Soucy *et al*, 2009; Fig 5D). A mutant of cyclin F lacking a functional F-box domain (ΔF) and unable to form and SCF complex interacted with E2F1 equally well in the presence or absence of MLN4924 (Fig 5D), indicating that cyclin F WT is promoting ubiquitylation of E2F1 but a mutant of cyclin F lacking catalytic activity (ΔF) is unable to

promote E2F1 ubiquitylation. We have previously shown that cyclin F interacts with its substrates using the cyclin domain and a mutant of cyclin F (M309A) in critical residues for the cyclin domain folding is unable to do so (D'Angiolella *et al*, 2010, 2012). The interaction between cyclin F and E2F1 was abolished by a single point mutation in the cyclin domain of cyclin F (Fig 5D). In accordance with the interaction data, cyclin F WT promotes the ubiquitylation of E2F1, while cyclin F ΔF and M309A mutants were not promoting E2F1 ubiquitylation (Fig 5E). The difference in ubiquitylation of E2F1 was not due to different levels of expression of cyclin F and E2F1 which were comparable in the input (Fig EV4C). Furthermore, the half-life of E2F1 was increased in *CCNF* K/O cells (Fig EV4D), providing strong evidence that E2F1 is a novel ubiquitylation substrate of cyclin F.

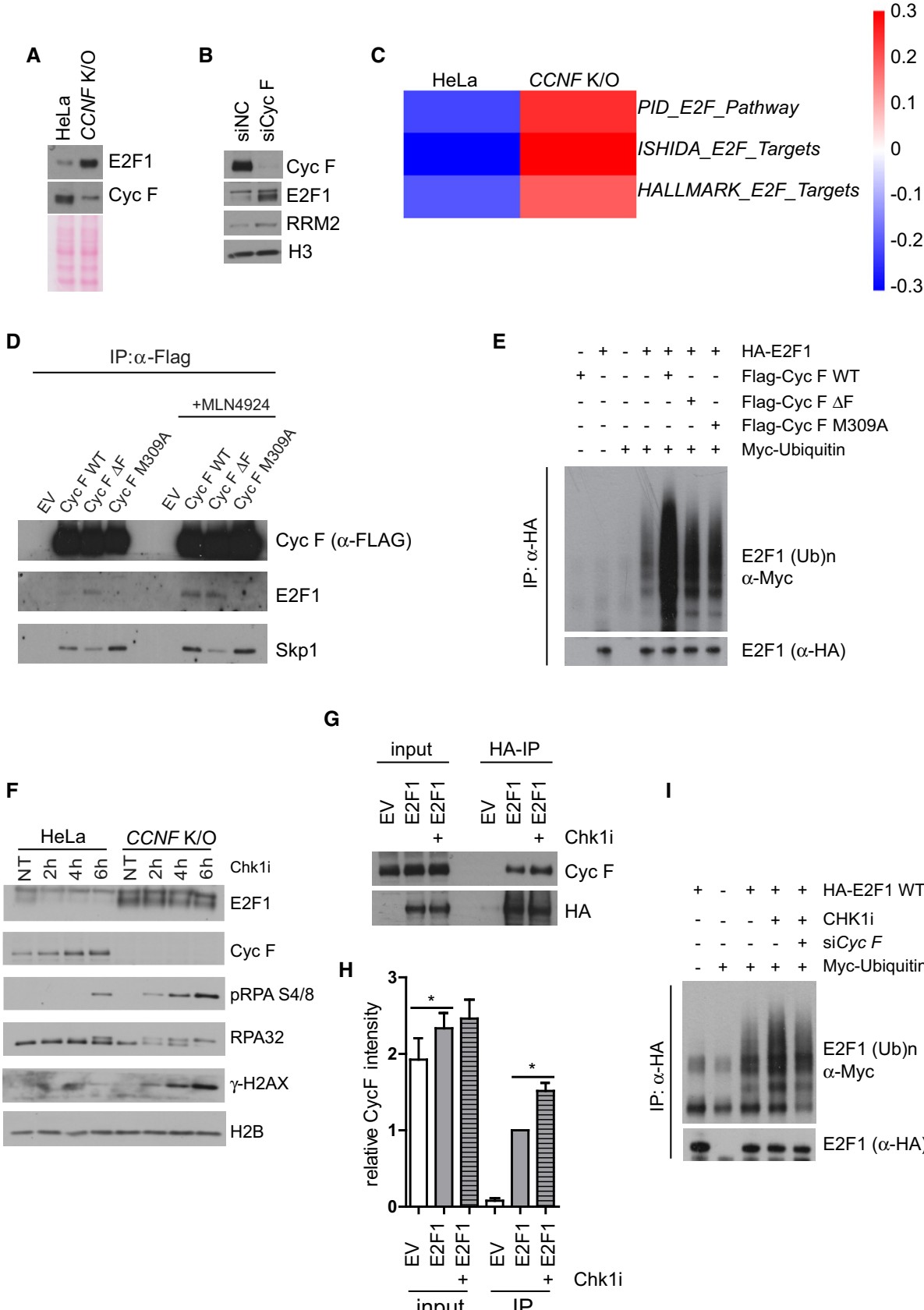

Figure 5.

**Figure 5. E2F1 is a ubiquitylation substrate of SCF^cyclin F degraded at the G2/M transition and after checkpoint inhibitors.**

A   HeLa and *CCNF* K/O cells were harvested and lysed using SDS. The indicated proteins were resolved by SDS–PAGE and detected by WB. Ponceau S staining was used as a loading control.

B   U-2-OS cells transfected with non-targeting siRNA (siNC) or with an siRNA targeting Cyc F were lysed using SDS 48h after transfection. Whole cell lysates were analysed by immunoblotting. H3 was used as a loading control.

C   Heatmap showing gene set variance analysis (GSVA) score for gene sets related to E2F targets (PID_E2F_PATHWAY, ISHIDA_E2F_TARGETS and HALLMARK_E2F_TARGETS).

D   HEK293T cells transfected with empty vector (EV), FLAG-cyclin F WT, FLAG-cyclin F (ΔF) and FLAG-cyclin F M309A were treated with MLN4924 for 4 h and harvested for immunoprecipitation using FLAG antibodies. Immunoprecipitates were immunoblotted as indicated.

E   HEK293T cells were cotransfected with MYC-tagged ubiquitin and HA-E2F1 in the presence of FLAG-cyclin F WT, FLAG-cyclin F (ΔF) and FLAG-cyclin F M309A as indicated (+). HA-E2F1 was immunoprecipitated with an anti-HA resin under denaturing conditions and immunoblotted.

F   HeLa and *CCNF* K/O cells treated with 10 μM Chk1i (LY2603618) for the indicated time points (h = hours) were harvested and lysed using SDS. Indicated proteins were resolved by SDS–PAGE and detected by WB. H2B served as a loading control.

G   HEK293 cells transfected with empty vector (EV) or HA-E2F1 were treated with Chk1i for 20 h and harvested for immunoprecipitation using HA antibodies. Immunoprecipitates were immunoblotted as indicated.

H   Quantification of immunoprecipitation results of (G) from three independent experiments with mean % ± SD. *< 0.05 *P*-value and calculated by two-tailed paired *t*-test.

I   HEK293T cells were cotransfected with MYC-tagged ubiquitin and HA-E2F1 and then treated with Chk1i for 20 h and siRNA of cyclin F as indicated (+). HA-E2F1 was immunoprecipitated with an anti-HA resin under denaturing conditions and immunoblotted.

It was previously shown that E2F1 is degraded upon challenging cells with ATR inhibitors (Buisson *et al*, 2015); therefore, we tested whether E2F1 was degraded upon treatment with Chk1 inhibitors as well, and if the degradation was dependent on cyclin F. E2F1 is down-regulated upon challenging cells with Chk1i (Fig EV4E). E2F1 down-regulation was prevented by treating cells with MG-132 or MLN4924 (Fig EV4E). In *CCNF* K/O cells, E2F1 degradation was not triggered by Chk1i (Fig EV4E). While the degradation of E2F1 was quickly elicited in HeLa parental cells upon Chk1i, *CCNF* K/O cells started with high levels of E2F1, which remained unaffected over the course of the experiment (Fig 5F). The results above indicated that cyclin F is controlling degradation of E2F1 upon Chk1 inhibition. To establish that the effect of cyclin F on E2F1 upon Chk1 inhibition was direct, we measured interaction between cyclin F and E2F1 upon treating cells with Chk1 inhibitors. The interaction between cyclin F and E2F1 was increased upon Chk1 inhibition (Fig 5G and quantified in H). Furthermore, ubiquitylation of E2F1 was triggered by treatment with Chk1 inhibitors (Fig 5I). The increased ubiquitylation of E2F1 observed upon Chk1 inhibition was reduced by concomitant depletion of cyclin F using siRNA (Fig 5I). Our findings reveal that E2F1 is a novel ubiquitylation substrate of cyclin F degraded at the G2/M transition and upon Chk1 inhibition.

**A mutant of E2F1 lacking a CY motif is not ubiquitylated by cyclin F and promotes DNA replication stress and cell death after Chk1 inhibitors**

Similarly to RRM2 and CP110, previously identified substrates of cyclin F (D'Angiolella *et al*, 2012), E2F1 is recruited by cyclin F using the cyclin domain, indicating the existence of a CY motif (RxL —cyclin binding domain) (Schulman *et al*, 1998) in E2F1 mediating the interaction with cyclin F. We identified an RxL at position 89–91 in E2F1 which promotes interaction with cyclin F; indeed, an E2F1 lacking this region failed to interact with cyclin F (Fig 6A). In accordance with the lack of interaction with cyclin F, E2F1 Δ RxL was also not ubiquitylated by cyclin F (Figs 6B and EV5A).

To test the stability of E2F1 and its effect on DNA replication stress, E2F1 WT and Δ RxL were exogenously expressed in cells under the control of a weak retroviral promoter (pBabe), which maintain expression of E2F1 to near-endogenous levels. In these

cells, when comparing the half-life of E2F1 WT and ΔRxL, we observed a significant increase of E2F1 ΔRxL half-life (Fig 6C and quantified in D). The increased stability of E2F1 ΔRxL corresponded to high levels of E2F1 ΔRxL during cell cycle progression. Indeed, while E2F1 WT was degraded in G2/M, E2F1 ΔRxL was not (Fig EV5B), in agreement with E2F1 ΔRxL not being targeted for degradation by cyclin F.

In addition to cell cycle regulation, Chk1 inhibitors triggered the degradation of the endogenous and exogenous E2F1, while a mutant of E2F1 lacking the CY motif was not degraded, demonstrating that the degradation of E2F1 is operated by cyclin F through the CY motif (Fig 6E). Most importantly, upon addition of Chk1 inhibitors there was a significant accumulation of DNA damage markers of ssDNA and DSBs, which was more prominent in cells expressing a non-degradable version of E2F1 (Fig 6E). Finally, the expression of E2F1 promoted more cell death upon Chk1 inhibition (Fig 6F). The cell death observed upon expression of E2F1 was significantly exacerbated by expressing a non-degradable version of E2F1, demonstrating that the stabilisation of E2F1 was promoting DNA replication stress and cell death upon treatment with Chk1 inhibitors (Fig 6F). Overall, the use of a non-degradable version of E2F1 (ΔRxL) solidly establishes that the DNA replication stress and cell death observed in *CCNF* K/O cells are mediated by accumulation of E2F1.

## Discussion

Here, starting from a drug screen, we identify a novel synthetic lethal interaction between cyclin F loss and Chk1 inhibition. Our study highlights that *CCNF* K/O cells, upon Chk1 inhibition, undergo DNA replication catastrophe, which is followed by cell death. We observe that synthetic lethality is promoted by the accumulation of E2F1, present in cyclin F-depleted cells. The concomitant depletion of E2F1 prevents DNA replication catastrophe in *CCNF* K/O cells. We further show that E2F1 is ubiquitylated by cyclin F during cell cycle progression and upon Chk1 inhibition. Using a non-degradable E2F1 (ΔRxL), we recapitulate increased DNA damage and cell death upon Chk1 inhibition. We propose a model where cyclin F can restrict E2F1 activity by controlling its levels when cells are treated with Chk1 inhibitors. The degradation of E2F1 operated by cyclin F

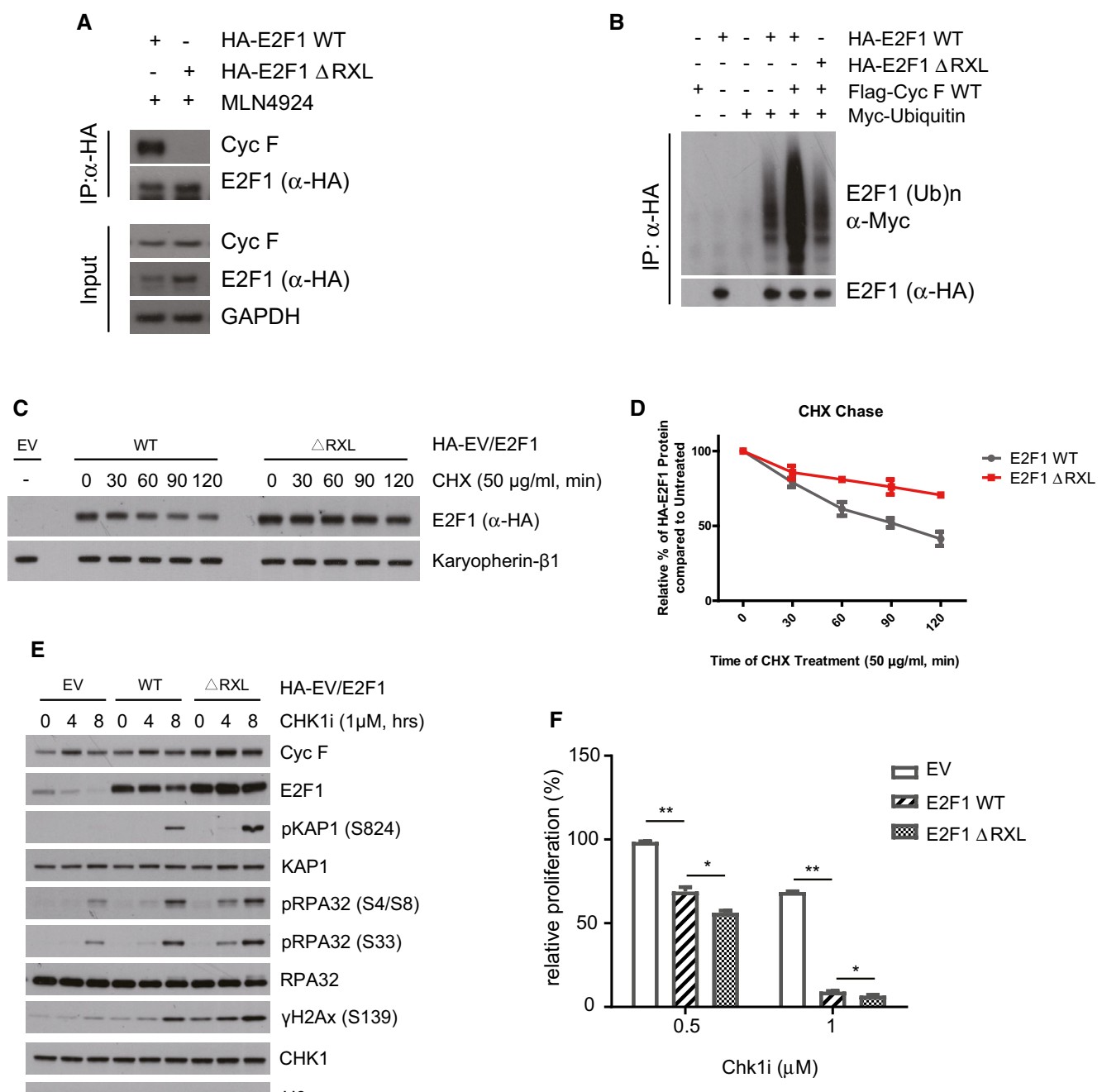

**Figure 6. A mutant of E2F1 lacking a CY motif is not ubiquitylated by cyclin F and promotes DNA replication stress and cell death after Chk1 inhibitors.**

A   HEK293T cells transfected with HA-E2F1 wild type (WT) and HA-E2F1 lacking a CY motif (ΔRxL) were harvested for immunoprecipitation using HA antibodies. Immunoprecipitates were immunoblotted as indicated.

B   HEK293T cells were cotransfected with MYC-tagged ubiquitin and HA-E2F1 or HA-E2F1 ΔRxL lacking the CY motif as indicated (+). HA-E2F1 was immunoprecipitated with an anti-HA resin under denaturing conditions and immunoblotted.

C   HeLa cells stably expressing HA-E2F1 WT and HA-E2F1 ΔRxL under the control of a retroviral promoter were treated with cycloheximide (CHX) at 50 μg/ml for the indicated minutes (min).

D   Quantification of E2F1 half-life in (C).

E   HeLa cells stably expressing an empty vector (EV), HA-E2F1 WT and HA-E2F1 ΔRxL under the control of a retroviral promoter were treated with Chk1i for the indicated hours (h). Indicated proteins were resolved by SDS–PAGE and detected by WB. H3 was used as a loading control.

F   Relative survival (%) of HeLa cells stably expressing an empty vector (EV), HA-E2F1 WT and HA-E2F1 ΔRxL under the control of a retroviral promoter measured using resazurin and compared to EV treated with Chk1i at the indicated concentrations.

Data information: Data are presented as mean ± SD, with at least three independent experiments. P-values (*P < 0.05, **P < 0.005) were calculated by paired and two-tailed t-test.

suppresses DNA replication progression. The concomitant loss of cyclin F and Chk1 prompts uncontrolled E2F1 activity in the absence of checkpoint control inducing DNA replication catastrophe.

The expression of a non-degradable version of E2F1 recapitulated the phenotype of cyclin F depletion giving strong proof that the DNA replication stress promoted by loss of cyclin F upon Chk1 inhibitors is mostly mediated by E2F1 accumulation. However, the phenotype of cyclin F loss was more striking than the single deregulation of E2F1. While this could be also ascribed to technical issues related to the generation of *CCNF* K/O cell lines and the expression of E2F1 near to physiological levels, it is also possible to speculate that other cyclin F substrates could play an additive role in inducing DNA replication catastrophe. Interestingly, it has very recently been shown that cyclin F targets E2F1 and its paralogs E2F2 and E2F3 for degradation (Clijsters *et al*, 2019). While we have excluded E2F3 as a mediator of DNA replication catastrophe in *CCNF* K/O (Fig 4A and B), it is plausible that E2F2 accumulation could also play a role.

Our data are in accordance with the observation that Rb loss was shown to promote increase sensitivity to Chk1 inhibition due to deregulated DNA replication (Witkiewicz *et al*, 2018). The observed phenotype of increased DNA replication stress and accumulation of DNA damage upon Chk1 inhibition in Rb-deficient breast cancer cell lines (Witkiewicz *et al*, 2018) closely resembles the DNA replication catastrophe mediated by cyclin F loss and E2F1 deregulation.

The mechanism of DNA replication catastrophe induced by E2F1 accumulation in cyclin F-depleted cells could be seen as oncogene-induced DNA replication stress, mostly studied in models of cyclin E overexpression. It has been shown that cyclin E accumulation leads to reduced DNA replication origin assembly (Ekholm-Reed *et al*, 2004). Furthermore, cyclin E overexpression alters the normal DNA replication programme forcing the formation of intragenic origins of replication. This process predisposes cells to the deleterious consequences of transcription replication collisions and results in the formation of DSBs (Macheret & Halazonetis, 2018). It is possible that the accumulation of E2F1 in *CCNF* K/O cells also predisposes the cells to the deleterious consequences of cyclin E deregulation and contributes to the induction of DNA replication catastrophe observed upon Chk1 inhibition. However, there are also substantial differences, as cyclin E overexpression has not been connected to Chk1 inhibitor sensitivity. Recent evidence suggests that transcription and DNA replication initiation sites are closely linked (Chen *et al*, 2019). We found that cyclin F regulates E2F1 only in the G2/M phase of the cell cycle. Given the central role of E2F1 in the transcription of genes essential for DNA replication initiation, it is tantalising to speculate that cyclin F regulates a late DNA replication programme, which entails E2F1 depletion. It is worth pointing out that the DNA replication stress induced by cyclin F depletion does not rely on overexpression systems. Therefore, the *CCNF* K/O system of DNA replication catastrophe promoted by Chk1 inhibition could be a more physiological system compared to cyclin E overexpression to study deregulated DNA replication and warrants further investigation.

Chk1 inhibitors are being evaluated in clinical trials in a variety of solid tumours. Inactivating mutations in cyclin F detected as diploid truncating mutations have been identified in stomach adenocarcinoma, lung adenocarcinoma, medulloblastoma, squamous cell carcinoma of the head and neck, glioblastoma, mucinous adenocarcinoma of the colon and rectum, colon carcinoma and cutaneous melanoma (Cerami *et al*, 2012; Gao *et al*, 2013). In light of our observations, we believe that patients should be selected based on cyclin F status. Given the striking synthetic lethality we report, patients with cancers bearing truncating mutations in cyclin F could benefit significantly from treatment with Chk1 inhibitors.

Interestingly, our drug screen reveals a number of positive and negative drug interactions, which could be further investigated. Cyclin F is amplified in 10-15% of breast invasive ductal carcinoma (Cerami *et al*, 2012; Gao *et al*, 2013). It is likely that amplification of cyclin F hampers the effectiveness of Chk1 inhibitor treatment. In the future, it would be crucial to establish cancer models that mimic cyclin F amplification both *in vivo* and *in vitro*.

# Materials and Methods

### Cell lines

HEK293, U-2-OS, MCF7, HeLa and HeLa *CCNF* K/O cells were cultured in DMEM (Sigma) containing 10% FBS (Gibco) and 100 U/ml penicillin, 100 mg/ml streptomycin (Millipore). HEK293, U-2-OS and HeLa cell lines were purchased from the ATCC. *CCNF* K/O cells were generated previously from HeLa cells (Mavrommatis *et al*, 2018). MCF7 was obtained from the Chapman laboratory.

### Antibodies and chemicals

| Antibody | Company | Cat. No. |
| --- | --- | --- |
| CCNA | a kind gift of Michele Pagano | |
| CCNB | Invitrogen | AHF-0052 |
| CCNE1 | Merck Millipore | 05-363 |
| CCNF (C-20) | Santa Cruz | sc-952 |
| Cdc6 (180.2) | Santa Cruz | sc-9964 |
| Chk1 (2G1D5) | Cell Signalling Technologies | 2360S |
| E2F1 | Santa Cruz | sc-251 |
| E2F3 (PG37) | Santa Cruz | sc-69684 |
| Exo1 | GeneTex | GTX109891 |
| Geminin (FL-249) | Santa Cruz | sc-13015 |
| GFP | Roche | 11814-460-001 |
| γ-H2AX(Ser139) | Novus Biologicals | NB 100-2280 |
| H3 | Cell Signalling Technologies | 9715 |
| HA-HRP | Roche | 12013819001 |
| KAP1 | Abcam | ab10484 |
| MTBP (K20) | Santa Cruz | sc-47173 |
| p53 (DO-1) | Santa Cruz | sc-126 |
| p53R2 (C-18) | Santa Cruz | sc-10843 |
| pCDK1 (Thr14/Tyr15) | Santa Cruz | sc-12340 |
| pCHK1 S296 | Abcam | ab79758 |
| pCHK1 S317 | GeneTex | GTX22834 |
| PCNA (PC10) | Santa Cruz | sc-56 |
| pHistone H3 (Ser10) | Millipore | 06-570 |
| pKAP1 S824 | Abcam | ab70369 |

### Antibodies and chemicals (continued)

| Antibody | Company | Cat. No. |
|---|---|---|
| pp53 (Ser15) | Cell Signalling Technologies | 9284 |
| pRPA32(S4/S8) | Bethyl | A300-245A-T |
| pRPA32(ser33) | Bethyl | A300-246A-M |
| P-ser CDKs substrate | Cell Signalling Technologies | 9477S |
| RPA32 (9H8) | Abcam | ab-2175 |
| RRM1 (T-16) | Santa Cruz | sc-11733 |
| Skp1 P19 | Santa Cruz | sc-5281 |

| Chemicals | Company | Cat. No. |
|---|---|---|
| Cycloheximide (CHX) | Sigma | C7698 |
| DAPI | Thermo Fisher | 62248 |
| EdU | Sigma | 900584-50MG |
| Low melting agarose | Sigma-Aldrich | 16520050 |
| LY2603618 Chk1 inhibitor | Stratech Scientific Ltd | S2626-SEL-5 mg |
| MG132 | Sigma-Aldrich | C2211-5MG |
| MLN4924 | Merck Chemicals | 5.05477.0001 |
| Nocodazole | MP Biomedicals | M1404-5MG |
| SYBR Gold | Thermo Fisher | S-11494 |
| Thymidine | Sigma-Aldrich | T9250-5G |
| UCN-01 Chk1 inhibitor | Cambridge Bioscience | 18130-1 mg-CAY |
| VE-821 ATR inhibitor | Stratech Scientific Ltd | S8007 |

### Sample preparation: trypsin digestion

Cyclin F was eluted from the GFP beads using 10 mM Tris, 2% SDS. The cyclin F eluted fractions were sequentially incubated with DTT (5 mM final concentration) and iodoacetamide (20 mM final concentration) for 30 min each at room temperature in the dark, before the proteins were precipitated twice with methanol/chloroform to remove SDS. Protein precipitates were reconstituted and denatured with 8 M urea in 20 mM HEPES (pH 8). Samples were then further diluted to a final urea concentration of 1 M using 20 mM HEPES (pH 8.0) before adding immobilised trypsin for 16 h at 37°C (Pierce 20230). Trypsin digestion was stopped by adding TFA (final concentration of 1%) and trypsin removed by centrifugation. Tryptic peptide mixture was desalted using the SOLA HRP SPE cartridges (Thermo Fischer) and dry down.

### LC-MS/MS and data analysis

Dried tryptic peptides were reconstituted in fifteen microlitres of LC-MS grade water containing 1% acetonitrile and 0.1% TFA. Thirty-three per cent of the sample was analysed by liquid chromatography-tandem mass spectrometry (LC-MS/MS) using a Dionex Ultimate 3000 UPLC coupled to a Q-Exactive HF mass spectrometer (Thermo Fisher Scientific). Peptides were loaded onto a trap column (PepMapC18; 300 μm × 5 mm, 5 μm particle size, Thermo Fischer)

for 1 min at a flow rate of 20 μl/min before being chromatographic separated on a 50 cm-long Easy-Spray column (ES803, Thermo Fischer) with a gradient of 2–35% acetonitrile in 0.1% formic acid and 5% DMSO with a 250 nl/min flow rate for 60 min. The Q-Exactive HF was operated in a data-dependent acquisition (DDA) mode to automatically switch between full MS-scan and MS/MS acquisition. Survey-full MS scans were acquired in the Orbitrap mass analyser over an $m/z$ window of 375–1,500 and at a resolution of 60k (AGC target at 3e6 ions). Prior to MSMS acquisition, the top twelve most intense precursor ions (charge state $\geq$ 2) were sequentially isolated in the Quad ($m/z$ 1.2 window) and fragmented on the HCD cell (normalised collision energy of 28). MS/MS data were obtained in the orbitrap at a resolution of 30,000 with a maximum acquisition time of 45 ms, an AGC target of 5e5 and a dynamic exclusion of 27 s.

The raw data were searched against the Human UniProt–SwissProt database (June 2018; containing 20,361 human sequences) using Mascot data search engine. The search was carried out by enabling the Decoy function, while selecting trypsin as enzyme (allowing 1 missed cleavage), peptide charge of +2, +3, +4 ions, peptide tolerance of 10 ppm and MS/MS of 0.05 Da; #13C at 1; carbamidomethyl (C) as fixed modification; and deamidated (NQ), oxidation (M), phospho (ST), phospho (Y) as a variable modification. MASCOT outputs were filtered using an ion score cut off of 20 and a false discovery rate (FDR) of 1%. A qualitative analysis was carried on with proteins identified with two or more peptides. LC-MS/MS raw data could be made available by the authors upon request.

### Kinase and drug screen

HeLa and *CCNF* K/O cells were seeded at 1,000 cells per well in 384-well plates 1 day before treatment with the KCGS library at 10, 1 and 0.1 μM concentrations. Relative cell survival was determined using resazurin 3 days after treatment. *Z*-scores were calculated for all samples, and the difference between HeLa versus *CCNF* K/O cells was plotted.

### Relative cell proliferation assay

U-2-OS cells were seeded at 2,000 cells per well to 96-well plate in hexaplicate 1 day before siRNA transfection. Cells were treated 1 day after siRNA transfection with indicated concentrations of ATRi or Chk1i. HeLa and *CCNF* K/O cells were seeded at 2,000 cells per well to 96-well plate in hexaplicate 1 day before treatment with indicated concentrations of ATRi or Chk1i. Relative cell proliferation was determined 3 days after treatment using resazurin 30 μg/ml in complete media after 1- to 2-h incubation.

### Cell counting and survival assay

U-2-OS cells were seeded at 20,000 cells per well to 12-well plate 1 day before transfection with siRNA. Cells were treated 1 day after siRNA transfection with indicated concentrations of Chk1i and analysed 3 days later. HeLa and *CCNF* K/O cells were seeded at 20,000 cells per well to 12-well plate 1 day before treatment with treatment with 1 μM Chk1i. Cells were counted, viability was determined using PI staining at 0.2 μg/ml 24, 48 and 72 h by FACS Attune (Life Technologies), and data were analysed using FlowJo software (TreeStar).

## Flow cytometry of chromatin bound proteins, cell cycle analysis and EdU staining

To analyse cell cycle profile, cells were pulse-labelled with 10 μM EdU for 30 min, harvested by trypsinisation and fixed in 70% ethanol. For analysis of chromatin bound proteins, cells were pulse-labelled with 10 μM EdU for 30 min where indicated, harvested by trypsinisation, washed in PBS, pre-extracted in pre-extraction buffer (25 mM HEPES pH 7.4, 50 mM NaCl, 1 mM EDTA, 3 mM MgCl$_2$, 0.3 M sucrose, 0.5% Triton X-100) 5 min on ice, centrifuged and fixed with 4% PFA 15 min at room temperature (RT). Fixed cell samples were permeabilised with 0.5% Triton X-100 for 15 min and blocked in 1% BSA. Click reaction was performed in buffer containing 0.1 M Tris–HCl pH 8.5, 0.1 M sodium ascorbate, 2 mM CuSO$_4$, 10 μM Alexa 647-azide 30 min at room temperature. Cells were washed in PBS and incubated with primary antibodies for 2 h at RT before incubation with secondary antibodies 1 h at RT. Cells were resuspended in DAPI 5 μg/ml in PBS, and data were acquired using FACS Attune (Life Technologies) and analysed using FlowJo (TreeStar).

## Immunoprecipitation and immunoblotting

HEK293 cells were seeded to 10-cm plates 1 day before plasmid transfection (5 μg/plate using PEI-MAX 40 kDA at ratio 6:1). After 20 h, cells were treated with 1 μM MLN4924 or 1 μM Chk1i for 4–5 h. Cells were washed twice with PBS, harvested to lysis buffer (50 mM Tris pH 7.5, 150 mM NaCl, 10 mM β-glycerophosphate, 10% glycerol, 1% Tween-20, 0.1% NP-40) containing protease inhibitors (Sigma) and okadaic acid, incubated on ice for 30 min and centrifuged for 30 min at 20,000 *g* 4C. Supernatant was incubated at 4C with either Flag M2 beads or HA beads for 1 and 2 h, respectively, before washing four times with lysis buffer. Immunoprecipitates were eluted using 2× LDS buffer (Life Technologies) supplemented with β-mercaptoethanol and boiled for 5 min at 95°C. Whole cell lysates were harvested using 2xSB (62.5 mM Bis-Tris pH 6.8, 20% (w/v) glycerol, 4% (w/v) SDS), boiled and sonicated, and protein concentration was evaluated using BCA protein kit (Thermo) in most cases. Cell lysate or immunoprecipitate was resolved in 10% Bis-Tris gels, transferred to nitrocellulose membrane (Millipore) and immunoblotted.

## DNA constructs, siRNA and transfections

CCNF gene was subcloned from Flag-cyclin F (D'Angiolella *et al*, 2012) into pYFP-C1. The product YFP-cyclin F was fully sequenced verified. HA-E2F1 plasmid was a kind gift of Nick La Thangue. Plasmid transfections were carried using PEI (PEI-MAX 40kDA, Polysciences) at ratio 6:1.

All siRNAs were purchased from Ambion: NC (4390843), CCNF (s2527, s2528, s2529), cdc6 (s2744), Cdt1 (s37724), Geminin (s27307), MTBP (s25788), TICRR (s40361), E2F3 (s4411), E2F1 (s4405, s4406), Exo1 (s17502) and transfected 10 nM. siRNA transfections were performed using HiPerFect (Qiagen) following manufacturer instructions.

## Comet assay

Alkaline and neutral comet assays were performed in parallel on the same cell set. Per condition, 50,000 trypsinised cells were mixed with 1% low melting point agarose (37°C) and added to microscope slides pre-coated with 1% normal melting point agarose. For the alkaline assay, cells were lysed for 1 h at 4°C in a buffer containing 2.5 M NaCl, 100 mM EDTA, 10 mM Tris–HCl, 1% (*v/v*) DMSO and 1% (*v/v*) Triton X-100, pH 10.5. Slides were placed in an electrophoresis tank and covered with 4°C electrophoresis buffer (300 mM NaOH, 1 mM EDTA, 1% (*v/v*) DMSO, pH > 13) for 30 m, to allow DNA to unwind. Electrophoresis was performed at 25 V for 25 m at 300 mA. Slides were neutralised in 0.5 M Tris–HCl pH 8.1 and allowed to dry overnight. For the neutral assay, cells were lysed for 2 h at 4°C in a buffer containing 2.5 M NaCl, 100 mM EDTA, 10 mM Tris–HCl, 1% (*v/v*) sodium lauroyl sarcosinate, 10% (*v/v*) DMSO and 0.5% (*v/v*) Triton X-100, pH 9.5. Slides were washed in TBE buffer (4°C), and electrophoresis was performed in TBE buffer (4°C) for 25 m at 25 V. Slides were allowed to dry overnight. For both assays, cells were rehydrated in H$_2$O, stained with SYBR Gold for 30 m and allowed to dry. Analysis was performed using the Nikon NiE microscope and Andor Komet 7.1 software. Two microscope slides were analysed per condition, with at least 100 cells per condition per biological replicate. Data were expressed as Olive tail moment: (Tail.mean − Head.mean)*Tail% DNA/100. *P*-values were calculated using the two-tailed Mann–Whitney test.

## RNA sequencing

800,000 cells were seeded into 10-cm dish in triplicate for each cell line. The next day, cells were around 80% confluence and had no uneven distribution. Dishes were then taken out from incubator and proceeded to cell lysing one at a time. For lysing cells, cells were quickly washed with 5 ml PBS twice and transferred immediately onto dry ice to flash-freeze. After transferring the frozen dish onto ice, 600 μl lysing solution (buffer RLT) from the RNA extraction kit was added and quickly spread all over the dish. Cell lysates were then collected into a 1.5-ml Eppendorf tube and stored on ice. Once collected and lysed all samples, all lysates were equilibrated to room temperature and proceeded to subsequent RNA extraction according to the kit manual (RNeasy Mini Kit from Qiagen, 74106). Sample concentrations, 260/280 and 260/230 ratios, were measured via NanoDrop as a quality control before being diluted to 200 ng/μl and stored in −80°C. ERCC ExFold RNA spike-in mixes (Ambion) were added prior to library preparation using the QuantSeq Forward Kit (Lexogen, 015.96) using 500 ng of starting material to minimise the PCR amplification step. Samples prepared as biological triplicates were sequenced on HiSeq 2500 (Illumina) carried out using Wellcome Trust Genomic Service, Oxford.

## RNA-seq data analysis

The quality control of raw sequencing reads was performed using FastQC (www.bioinformatics.babraham.ac.uk/projects/fastqc). Raw fastqc files were trimmed of poly-A using Cutadapt (Martin, 2011). All trimmed reads were aligned to the hg19 genomic reference using STAR software (Dobin *et al*, 2013). Counts per gene from STAR were used as input for differential gene expression analysis using Limma (Ritchie *et al*, 2015).

### Gene set enrichment analysis

Gene set enrichment analysis (GSEA) analyses were carried out using GSEA software (javaGSEA 3.0) (Subramanian et al, 2005) with 1,000 permutations and default parameters using PID_E2F_PATHWAY, ISHIDA_E2F_TARGETS and HALLMARK_E2F_TARGETS gene signatures. Gene set variation analysis (GSVA) analyses were performed using the GSVA package using same E2F gene signatures (Hanzelmann et al, 2013). The gene sets were obtained from the Molecular Signatures Database (MSigDB; http://software.broadinstitute.org/gsea/msigdb/index.jsp).

### *In vivo* ubiquitylation assay

HEK293T cells were seeded into 10-cm dishes. When reaching 60% confluency, cells were transfected with plasmids as indicated in each experiment (for a 10-cm dish, 1 µg HA-E2F1, 2 µg Flag-cyclin F or 5 µg myc-ubiquitin was transfected or cotransfected). Thirty hours after transfection, cells were harvested via scraping and washed with PBS once. Cell pellet from each dish was then thoroughly lysed and boiled in 300 µl ubiquitin lysis buffer (2% SDS, 150 mM NaCl, 10 mM Tris–HCl pH 7.4, supplemented with protease inhibitors). After cooling down to room temperature, cell lysates were then subjected to sonication until losing viscosity. Lysates were then boiled again and centrifuged at 20,000 $g$ for 10 min. 20 µl supernatant was preserved as input for each sample. The rest supernatant was diluted 20 times via dilution buffer (10 mM Tris–HCl pH 7.4, 150 mM NaCl, 2 mM EDTA, 1% Triton x-100) and proceeded to immunoprecipitation with HA beads (E6779-1ML, Sigma-Aldrich, 10 µl beads per sample). After 3 h of incubation on roller, beads were collected via centrifugation at 500 $g$ for 30 s and washed with 1 ml wash buffer (10 mM Tris–HCl pH 7.4, 1 M NaCl, 1 mM EDTA, 1% NP-40) for five times before being mixed with 50 µl 1× loading buffer, boiled and subjected to WB.

### Cell cycle synchronisation via double thymidine release (DTR)

300,000 cells were seeded into each 10-cm dishes (seven dishes per cell line). For HeLa and *CCNF* K/O cell lines, 24 h later seeding, cells were subjected to the first thymidine block (2 mM final concentration) for 16 h. Cells were then washed three times with PBS and once with full media before being released into fresh complete media for 8 h. A second thymidine block was then performed via the same condition. Sixteen hours after, 0 time points were collected before the rest of the cells were washed and released into fresh complete media. After releasing cells from the second thymidine block, cells were collected at different time points as indicated. At 7-h time point, 200 nM Nocodazole was added to the uncollected cells to prevent cells from entering the next cell cycle.

### Cycloheximide (CHX) chase

250,000 cells were seeded into each well of a 6-well plate (5 wells per cell line). Sixteen hours after, cycloheximide was added to cell culture as indicated to a final concentration of 50 µg/ml to block translation. Cells were collected via scraping and subjected to straight WB for protein half-life estimation.

### Statistical analysis

To evaluate statistical significance, $P$-value was determined using GraphPad Prism. Paired two-tailed $t$-test was used for relative survival assays. Two-way ANOVA test was used to evaluate statistical significance of cell proliferation in cell counting assay. Unpaired two-tailed $t$-test was used to evaluate statistical significance of percentage of γ-H2AX and hyper-RPA-positive cells. $P$-values are marked by asterisks in figures as follows: *$P < 0.05$, **$P < 0.005$, ***$P < 0.0005$.

### Image plotting in Fig 2H

The RPA2 against γ-H2AX dot plots in Fig 2H was originally generated using FlowJo V10 software, in which γ-H2AX-positive events were colour in blue, whereas the rest are in grey. The original $X$-axis (RPA2 intensity) and $Y$-axis (γ-H2AX) of each plot were removed and replaced with common RPA2 and γ-H2AX axes. Plots from different time points were then assembled against time axes in a staggered offset manner using Adobe Illustrator. Please refer to Fig EV2D for original data from the same experiment.

## Data availability

All raw data generated in this study are collected in Datasets EV1 and EV2. Data from the drug screen of *CCNF* K/O lines are uploaded as Dataset EV1 and summarised in Table 1. RNA-seq data are collected in Dataset EV2 and have also been deposited in the European Nucleotide Archive (ENA; https://www.ebi.ac.uk/ena/submit/sra/#studies) with primary accession number PRJEB33738 and secondary accession number ERP116555.

Expanded View for this article is available online.

### Acknowledgements

This study was possible thanks to the support of a Medical Research Council (MRC) Grant MC_UU_00001/7 to V. D'A. This work was supported by Cancer Research UK (CR-UK) Grant number C5255/A18085, through the Cancer Research UK Oxford Centre. This work was further supported by a John Fell (133/075) and Wellcome Trust grant (097813/Z/11/Z) to B.M.K. Funding for the SGC-UNC was provided by The Eshelman Institute for Innovation, UNC Lineberger Comprehensive Cancer Center, PharmAlliance and National Institutes of Health (1R44TR001916-02, 1R01CA218442-01 and U24DK116204-01). Mass spectrometry analysis was performed in the Discovery Proteomics Facility (headed by Roman Fischer) which is part of the TDI MS Laboratory (led by Benedikt Kessler).

### Author contributions

KB and HY designed the experiments, acquired, analysed and interpret data. RF contributed to the investigation and analysis of data with ATR inhibitors. SH and GD contributed to the investigation and analysis of comet assays. DE, DHD, CIW and SBH helped with the execution and analysis of KCGS screening data. BMK and IV ran and analyzed LC-MS/MS. JC and FMB conducted the RNA-seq analysis. VD'A wrote the manuscript, analysed results and coordinated the study.

### Conflict of interest

The authors declare that they have no conflict of interest.

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
