## [Review Process File · The EMBO Journal]

E2F1 proteolysis via SCF-Cyclin F underlies synthetic lethality between Cyclin F loss and Chk1 inhibition

Kamila Burdova, Hongbin Yang, Roberta Faedda, Samuel Hume, Jagat Chauhan, Daniel Ebner, Benedikt M Kessler, Iolanda Vendrell, David H Drewry, Carrow I Wells, Stephanie B Hatch, Grigory Dianov, Francesca M Buffa, Vincenzo D'Angiolella.

Review timeline:

Submission date:	1 st January 2019
Pre-decision consultation:	20 th February 2019
Author response:	7 th March 2019
Editorial Decision:	20 th March 2019
Revision received:	5 th July 2019
Accepted:	29 th July 2019

Editor: Hartmut Vodermaier

Transaction Report:

Pre-decision consultation

20th February 2019

Please excuse the delay in getting back to you with a response on your manuscript on Cyclin F/Chk1/E2F1 interplay, but we have now finally received a full set of comments from three expert referees. As you will see, the referees are somewhat divided in their overall opinion, but in any case it is apparent that all of them share significant concerns related to both conceptual aspects and to experimental conclusiveness and depth of the analyses. In particular, they wonder about the (patho-)physiological relevance of the identified interactions, bring up plausible alternative explanations, and criticize that the investigation of cyclin F-dependent E2F1 ubiquitination and degradation remains currently very preliminary/superficial. In light of these substantive concerns, I am afraid we cannot consider the study a very strong candidate for an EMBO Journal article at the present stage. Nevertheless, I would like to give you an opportunity to carefully consider the referee reports and to send back a brief point-by-point response outlining whether/how you might be able to address the issues raised by the reviewers and extend the paper. Based on these tentative response (parts of which we may choose to share and discuss with our referees), we would then decide whether a major revision for The EMBO Journal would seem realistic and justified in this case, as well as whether the study might alternatively be revisable for some of our sister journals such as EMBO reports or Life Science Alliance. It would be great if you could get back to me with such a response at the beginning of the coming week.

REFeree REPORTS

Referee #1 (Report for Author)

Burdova et al.

Cyclin F-Chk1 synthetic lethality mediated by E2F1 degradation

Overview/Suymmary

The SCF family of E3 ubiquitin ligases play key roles in cell cycle control. SCF ligases bind to substrates via a family of adaptors termed F-box proteins. Cyclin F is the founding member of the F-box family and has been shown to regulate several cell cycle substrates. Two of the key studies highlighting the importance of Cyclin F in cell cycle control were performed by the lead author as a postdoc, where he elegantly demonstrated that Cyclin F controls the abundance and stability of CP110 and RRM2.

In this current study, the authors sought to determine what kinase inhibitors impaired proliferation specifically in Cyclin F knockout (KO) HeLa cells. They screened a set of ~200 kinase inhibitors and found that Cyclin F KO cells were especially sensitive to inhibition of Chk1, a kinase involved in the DNA damage response pathway. They show that Cyclin F KO cells are sensitive to Chk1 and ATR inhibition. They also suggest that this could be mediated by E2F1.

Unfortunately, there were several aspects of this paper that need improvement. First, while it seems clear that Cyclin F KO HeLa cells are sensitive to Chk1 inhibition, it is not clear why this is an important finding. Cyclin F is never lost in cancers, and it is unclear what downstream signaling pathway would or could be elucidated by testing sensitivity to kinase inhibition, especially in HeLa cells. Second, the authors state in the title and the abstract that E2F1 is a Cyclin F substrate. However, they are far from meeting the field standard for having shown this. And third, they say in the abstract that Cyclin F loss combined with Chk1 inhibition leads to replication catastrophe, although they have failed to show this. For these and other reasons, this manuscript is not ready for publication.

Major Points

1. The authors show that Cyclin F KO HeLa cells are sensitive to Chk1 inhibition. However, these drug studies are not performed in the sort of concrete way that would be satisfying to someone doing pharmacology. If this is going to be a drug study, which is what the first half of this manuscript is, then they should be performing careful dose response curves over a large dose range and analyzing cell proliferation/numbers/etc. They approach this in Figure 1D, although these data are usually shown on a log scale. If the drug experiment was supportive of a finding in a study focusing on a mechanistic aspect of cell or molecular biology, this request would be unnecessary. But in this case, the first half of the manuscripts aims to showcase the sensitivity of Cyclin F KO cells to these drugs, and therefore, it should be done appropriately.
2. The authors fail to report the full data set from the drug screen, and thus, one cannot assess whether the Chk1 inhibitor that scored is an outlier among other inhibitors to the same target. The full data set would need to be reported to appreciate the importance of the screen.
3. As stated above, it is not clear why the authors are screening for vulnerabilities to Cyclin F loss. And why are they doing so in HeLa cells?
4. Figure 2K is an essential piece of data, showing that drug sensitivity is lost when Cyclin F is re-introduced. In its current form it is not convincing. The authors should have shown a blot for Cyclin F, demonstrating that it was restored to near endogenous levels. Moreover, they need to demonstrate that this eliminates the drug sensitivities observed in Figure 1, and not just alters the degree of DNA damage.
5. They indicate that IP-MS was done on Cyclin F, and that this enriched for proteins involved in cell cycle. These data would need to be shown to interpret this suggestion/implication. (Related to Figure 3A).
6. A simple explanation for the rescue of DNA damage in E2F1 depleted cells would be that they are now arrested in G1. This is not addressed.
7. Importantly, the authors don't show that depleting E2F1 rescues the sensitivity to Chk1 inhibitors, but instead, that it alters the degree of DNA damage. This is key given the title of the paper.
8. In Figure 3E, E2F1 levels are identical between control and Cyclin F KO cells. Then in Figure 4A, they are hugely different (several fold). This is a major discrepancy.
9. The authors are far from having met the standard to show that E2F1 is a Cyclin F substrate. Missing experiments are many, and include, demonstration of E2F1 degradation upon Cyclin F overexpression, an assay for E2F1 ubiquitination, mapping of a degron on E2F1.
10. When does Cyclin F control E2F1 in the cell cycle?

11. The Chk1 inhibitor experiment in Figure 4F is not convincing, since the Cyclin F depleted cells have much more DNA damage, which is presumably driving E2F1 degradation. Moreover, it is not at all evident that the kinetics of E2F1 loss is greater, since the levels start higher.
12. The suggestion that phosphorylation of E2F1 controls degradation is not sufficiently supported.

Minor Points

1. Figure 1B is lacking statistics.
2. The authors say that sensitivity is affected "more than several fold". This is imprecise.

Referee #2 (Report for Author)

In the manuscript entitled "Cyclin F-Chk1 synthetic lethality mediated by E2F1 degradation", the authors conducted a synthetic lethality (SL) screen in CCNF-KO background using kinase inhibitors. Here, they find a clear synthetic lethality relationship of CCNF-K/O with CHK1 inhibitors and to a minor extent ATR inhibitors. This combination of factors leads to a 'replication catastrophe', which intriguingly can be rescued by suppression of E2F1.

In general, the basic SL discovery between Cyclin F(CCNF) and CHK1i is exciting. However, aspects of the manuscript article are somewhat speculative with conclusions that are not sufficiently data-based. The overall data quality is fine, however, some restructuring would be an improvement. The corresponding author team is highly recognized in the ubiquitylation/CCNF field, and it is somewhat surprising that the CCNF-E2F1 interplay is explored in a rather limited fashion. Overall, the manuscript provides a significant contribution to the field, however, more detailed mechanistic investigations should be carried out to further substantiate conclusions.

---Major points:

*Figure 2 is very extensive and rather complicated to read. It would benefit from a more visibly clear way of presenting the data (2B, C, D), while some of the data might probably be better kept as supplementary (i.e. figure 2E, F, G) for clarity of the figure. Figure 2H+I are relatively more key experiments (than figure 2E, F, G) but need a better description and evaluation in the text. Also in general, the text description of this part does not make the data easier to read and understand.

*The authors are likely right regarding induction of replication catastrophe, however, they do not show this properly (for example by displaying RPA vs γ H2AX plots in FACS profiles to show the exhaustion of RPA).

*The presentation of the MS data and their further use in figure 3 is incomplete. It appears that a list (table) of CCNF interaction partners is missing, at least for the G/S factors suggested from figure 3A. Readers are left with unclear justification for choosing these factors for rescue experiments. In the absence of such table, it seemingly appears that (in Figure 3B) authors attempt phenotype rescue with knockdown of a very biased pathway-selected set of factors and not more of the known targets of CCNF. RRM2, SLBP and B-Myb might well contribute to the phenotype but they were not included (and should be).

*Figure 3E: siE2F1 also reduces CCNF levels. This indicates that siE2F1 might have considerable indirect effects on the cell cycle that dampen the phenotype following CHK1i. This notion should be tested by carefully analyzing the cell cycle and replication catastrophe +/-siE2F1 +/-CCNF +/-Chk1i.

*Role of E2F1 and/or downstream targets of E2F1: This aspect is not very developed. siE2F1 rescue might partly be mediated through reestablishing Cyclin E regulation though these data are less convincing. There is only one blot showing a partial rescue without good loading control. This could be removed or investigated properly including assays to understand CDK deregulation (also with assays in the model systems +/-CCNF +/- CHK1i). If the data are removed, it would be important to show that E2F1 targets are deregulated, which could be done at mRNA level or with E2F1 reporter (Cyclin E protein levels do not appear upregulated in CCNF K/O cells).

*The degradation mechanism should be further explored, which domains are required for the interaction in CCNF and E2F1, respectively? Complementation experiments with MR and F-BOX mutant CCNF are crucial and should give insight into how CCNF directly regulates E2F1. This is important in the absence of non-degradable E2F1 variant. The data in EV4 (A, D, and E) are very exciting and also discussed in the text in great detail, as such this should be a real figure, potentially figure 5 with the complementation data mentioned earlier. E2F1 half-life in EV4A should be quantified.

*The involvement of phosphorylation is not well developed and could be omitted (EV4B and C).

*SKP2 is a published ubiquitin ligase for E2F1, does SKP2 contribute to the observed regulatory circuitry? This question is obvious due to the lack of clear mechanistic data on the CCNF-E2F1 degradation.

---Additional points relating to the discussion:

*transcription and replication initiation site overlap aspects; explanations of this statement including references are missing for the role of E2F1 in the regulation of this process.

*In relation to clinical relevance, is CHK1 also displaying SL-interactions with RB-loss or Cyclin E amplification (in cells with wt-CCNF)?

*Claim of CCNF inactivation in cancer appears without support in the form of data and references. This should be provided.

Referee #3 (Report for Author)

In this manuscript the authors have screened for small molecule inhibitors synthetic lethal with Cyclin F. They identify Chk1 and ATR as targets whose inhibition kill HeLa and U2OS cells that lack Cyclin F and show that the synthetic lethality is due to DNA damage during S-phase. Further, they identify E2F1 as a target of Cyclin F, which is sensitive to inhibition of CDK activity. Depletion of E2F1 largely rescues the phenotype.

The data on synthetic lethality and DNA damage in S phase is strong and may be important from a clinical perspective. Further, it reinforces the notion that regulation of DNA replication is an important aspect of Cyclin F function - and by adding E2F1 hints of a mechanism of feedback between CDK activity and CDK-induced transcription, which has clear implications for genome stability and cell-cycle control. That E2F1 can interact with Cyclin F is clear, however, as E2F1 induces the transcription of other Cyclin F targets it is not clear whether that is the sole target for Cyclin F to counter replication stress.

Major points.

Only cancer cell lines are investigated. As in particular p53 has functional connections with both ATR-Chk1 and E2F1 it would be important to see if the main findings translate to non-transformed cells (p53 positive cancer cells are poor substitutes as pathways likely adapted).

It is important to test whether comparable populations are assessed after E2F1 siRNA. Does E2F1 siRNA affect cell-cycle progression, in particular amount of cells entering S-phase?

The manuscript is sparsely referenced throughout. For example, the paragraph describing clinical relevance in the discussion contains many examples but no references. The text is also not quite put into context: That Cyclin F has been described to downregulate cell cycle transcription through B-Myb is not described (Klein, Nat Com 2015), and that E2F1 has been described to be targeted for degradation by other factors is not mentioned (Martí, NCB 1999 and others).

I find figure 4G slightly confusing. The interactions leading to the arrows should be discussed/referenced. Why is CDK1 but not CDK2 included? Why is there no arrow between E2F1 and CDK activity? Especially as the authors argue that the function likely is partially through expression of Cyclins. Similarly, is the only interaction between CDK and E2F1 a negative one? At

the same time, Cyclin F can directly target E2F1 regulated genes that may impact replication. Absence of phenotype after depletion of E2F1 is therefore not conclusive for that E2F1 is the sole target of Cyclin F relevant for replication stress after Chk1 inhibition. This should be discussed and I'd suggest modifying figure 4G.

In discussion: "The mechanism of DNA replication catastrophe induced". First, maybe more accurate with "seen as similar to", rather than "seen as"? Second, to make this point it would be important to show replication stress in the absence of Chk1i. Although that may be selected against in stable knock-out clones, it should become apparent in depletion experiments.

To really make the point that Cyclin F targets E2F1 for degradation, it would be good to investigate whether Cyclin F supports ubiquitination of E2F1.

Minor points:

Fig EV4D, E. Are amount of mitotic cells similar? In particular did cells pass through mitosis in E?

I suppose the authors tried to test the interaction of endogenous Cyc F and E2F1? This should at least be commented upon.

"Cyclin F regulates E2F1 solely in the G2/M phases". I'm not sure I follow what data that is based on? Cyclin F expression from other publications? Please specify/reference. It may be helpful (maybe also in relation to point on 4G above) describing this in relation to CDK1 being restricted until S/G2 transition by Chk1 (Saldivar, Science 2018; Lemmens, Mol Cell 2018).

In figure 3E, is there no difference in E2F1 levels with or without Cyclin F? It would be helpful to include a lower exposure.

Fig 2 E,F. It would be helpful with examples of gating

I would find it helpful with some more data from the MS experiment as a figure.

Please include page and figure numbers to help reviewing.

OXFORD INSTITUTE FOR RADIATION ONCOLOGY

DEPARTMENT OF ONCOLOGY

Gray Laboratories, University of Oxford, Old Road Campus Research Building, Roosevelt Drive, Oxford OX3 7DQ

Vincenzo D'Angiolella

Tel : 01865617400

Email: vincenzo.dangiolella @oncology.ox.ac.uk

I believe we can address the reviewer comments and we provide some preliminary data that support the main findings of the manuscript.

In addition to what we propose below, we are also planning to gather more insights in the mechanisms of synthetic lethality by conducting DNA combing and DNA fiber assay experiments in *CCNF* K/O cells treated with Chk1i comparing them to HeLa parental.

The two major critiques from reviewers are:

1. Ubiquitylation of E2F1. Preliminary data are presented in Figure R1A and B

We added purified SCF^{cyclin F^{WT}}, SCF^{cyclin F^{M309A}} (lacking the cyclin domain and so interaction with E2F1) and SCF^{cyclin F^{ΔF}} (lacking the F-box domain and so activity) complexes to E2F1 and measured ubiquitylation of E2F1. Although data are preliminary it is clear that cyclin F WT is able to ubiquitylate E2F1 *in vitro* while the two other mutants are less efficient (Figure R1A and R1B). We are planning to optimise the assay during the revision and use an E2F1 lacking a cyclin F 'degron' identified in E2F1 at RxL 90-93 (Figure R2C).

2. E2F1 as a target in cyclin F-Chk1 synthetic lethality - S phase arrest.

We reanalysed the experiment presented in Figure 3 only considering cells in S phase to evaluate γ -H2AX levels by FACS. In S phase cells, the γ -H2AX levels and DNA damage can be reduced by removing E2F1 in the *CCNF* K/O treated with Chk1i (Figure R1C).

In addition, it is worth considering that although siRNA of Treslin and MTBP has been reported to arrest cells in S phase (Boos et al., 2013), siRNA of Treslin and MTBP does not reduce DNA damage in *CCNF* K/O cells exposed to Chk1i.

Finally, the experiments conducted in MCF7 with ATRi in Figure EV3H demonstrate a complete rescue of DNA damage. In the same settings we observe that the cell cycle distribution is not affected (Figure R1D). When measuring survival of MCF7 upon ATRi we observe that while cyclin F siRNA promotes cell death, this is fully abolished by siRNA of E2F1 (Figure R1E). The data presented above supports the idea that the DNA replication stress in *CCNF* K/O is mostly mediated by accumulation of E2F1 (oncogene induced DNA replication stress). We are planning to repeat similar experiments using an E2F1 mutant lacking the RxL 'degron' required for interaction with cyclin F.

Regarding the pathophysiological significance, it is clear to me that the study identifies a novel function of cyclin F, highlight a novel synthetic lethal interaction and provide a new model of DNA replication stress which does not rely on gene overexpression. Therefore, the significance is high and, indeed two reviewers out of three highlight this. However, we have further addressed your concerns on pathophysiological relevance in the reply point 3 to reviewer 1 below.

A point-to-point answer follows below:

Referee #1 (Report for Author)

We would like to thank the referee for reading carefully through our data and provide insightful comments which certainly will improve the manuscript. We agree that some aspects could be improved during the revision process and we are answering the reviewer concerns below:

1. The authors show that Cyclin F KO HeLa cells are sensitive to Chk1 inhibition. However, these drug studies are not performed in the sort of concrete way that would be satisfying to someone doing pharmacology. If this is going to be a drug study, which is what the first half of this manuscript is, then they should be performing careful dose response curves over a large dose range and analyzing cell proliferation/numbers/etc. They approach this in Figure 1D, although these data are usually shown on a log scale. If the drug experiment was supportive of a finding in a study focusing on a mechanistic aspect of cell or molecular biology, this request would be unnecessary. But in this case, the first half of the manuscripts aims to showcase the sensitivity of Cyclin F KO cells to these drugs, and therefore, it should be done appropriately.

We will conduct more dose response curves to support the synthetic lethal interaction between cyclin F and Chk1. Although it is possible that more time points and larger concentrations of drugs will not change the results, which were carefully taken. The data in Figure 1D will be shown on a log scale.

2. The authors fail to report the full data set from the drug screen, and thus, one cannot assess whether the Chk1 inhibitor that scored is an outlier among other inhibitors to the same target. The full data set would need to be reported to appreciate the importance of the screen.

We will upload as supplementary data the full data set. Please note two independent screens were conducted: one using the KCGS drug set and another one using a customised drug set of 350 oncology drugs. In the KCGS screen (for which full results are presented) we identified CCT244747. In the other screen we identified PF477736 and AZD7762 two Chk1 inhibitors. Among two screens conducted independently 3 Chk1 inhibitors appeared as top hits. Considering the data above Chk1 inhibitors are **not** outliers but the full data set is uploaded for clarity. We believed it was easier for readers and reviewers to visualise the results in the table 1. This is the sole reason we uploaded it in this fashion.

3. As stated above, it is not clear why the authors are screening for vulnerabilities to Cyclin F loss. And why are they doing so in HeLa cells?

Cyclin F is a crucial regulator of cell cycle progression and genome stability. Furthermore, mutations in cyclin F have been found in neurodegenerative diseases (Williams et al., 2016). Identifying

vulnerabilities could help basic cell biologists dissect cyclin F functions in various biological settings revealing novel pathways where cyclin F mediated ubiquitylation is crucial.

The manuscript is not intended to provide clinical significance but rather provide new insights into the role of cyclin F and Chk1. The discovery that cyclin F targets E2F1 and is synthetic lethal with Chk1 is a fundamental biological discovery which could have implications in the clinical settings for the reasons outlined below.

Cancers with diploid truncating mutations in cyclin F have been reported and studies are deposited in cbiportal (<https://www.cbiportal.org>):

In these cancers, it is possible to state that cyclin F is lost. The overall somatic mutation frequency across cancers is 1.1 %, it remains to be assessed whether mutations in cyclin F impact its function. In addition to mutations, cyclin F mRNA and protein levels as well as activity could vary. Please note it was reported that cyclin F activity is controlled by AKT phosphorylation (Choudhury et al., 2017). AKT is a known cancer driver. Finally, amplifications of cyclin F have been reported in more than 3 % of breast cancers (<https://www.cbiportal.org>).

I have the opinion that targeting sporadic events (like cyclin F mutations) could elicit a durable response as the emergence of these mutations in cancer is already a process of selection. It remains to be assessed whether targeting tumors lacking cyclin F in the clinic will make a difference, but this is well beyond the scope of this manuscript.

4. Figure 2K is an essential piece of data, showing that drug sensitivity is lost when Cyclin F is re-introduced. In its current form it is not convincing. The authors should have shown a blot for Cyclin F, demonstrating that it was restored to near endogenous levels. Moreover, they need to demonstrate that this eliminates the drug sensitivities observed in Figure 1, and not just alters the degree of DNA damage.

We have tried to reintroduce cyclin F in these cell lines with different retroviral systems and, usually the levels of cyclin F are lower than endogenous. For this reason, we have transiently expressed GFP-cyclin F and measured γ -H2AX in GFP positive cells with low cyclin F expression. The levels of cyclin F are higher than endogenous, however, the same phenotype of drug sensitivity is obtained in U2OS and MCF7 upon siRNA of cyclin F, demonstrating that this is not an off-target effect of the sgRNA and not a clonal effect of the CRISPR line.

We will attempt to repeat the suggested experiments using an inducible system tet-off system of cyclin F as previously done (D'Angiolella et al., 2012).

5. They indicate that IP-MS was done on Cyclin F, and that this enriched for proteins involved in cell cycle. These data would need to be shown to interpret this suggestion/implication. (Related to Figure 3A).

For simplicity we have highlighted the gene enrichment sets. We have selected interacting partners that were previously involved in DNA replication stress and sensitivity to Chk1i. In the mass spectrometry we identified E2F3, however loss of RB was previously shown to sensitise cells to Chk1i through E2F1 accumulation (Witkiewicz et al., 2018). Therefore, in our small screen we selected the highlighted candidates. The discussion pertaining to the screen conducted in Figure 3A and B will be expanded and clarified during the revision. A table summarising the mass spec scores of the candidates identified and used for the screen will be provided and added to the figure.

6. A simple explanation for the rescue of DNA damage in E2F1 depleted cells would be that they are now arrested in G1. This is not addressed.

To exclude that this is only due to cell cycle arrest we reanalysed the experiment in Figure 3 to determine the levels of γ -H2AX only in S Phase cells based on DNA content. When analysing only S phase cells it is clear that siRNA of E2F1 suppresses the formation of γ -H2AX (Figure R1C).

In addition, it is worth considering that although siRNA of Treslin and MTBP has been reported to arrest cells in S phase (Boos et al., 2013), siRNA of Treslin and MTBP does not reduce DNA damage in *CCNF* K/O cells exposed to Chk1i.

Furthermore, challenging MCF7 cells with ATR inhibitors and combined siRNA of cyclin F and siRNA of E2F1 rescue the DNA damage defect (Figure EV3H and I) without impacting on cell cycle distribution (Figure R1D). The combined siRNA of cyclin F and siRNA of E2F1 in MCF7 treated with ATR inhibitors fully rescued cell viability (Figure R1E).

At the same time, we can't exclude the fact that the accumulation of E2F1 accelerates entry into S phase, which could be part of the mechanism of synthetic lethality. We believe that expressing a mutant of E2F1 lacking the 'degron' and testing the synthetic lethality using this mutant could further clarify this point.

7. Importantly, the authors don't show that depleting E2F1 rescues the sensitivity to Chk1 inhibitors, but instead, that it alters the degree of DNA damage. This is key given the title of the paper.

The viability of HeLa cells upon siRNA of E2F1 and Chk1 inhibition is not fully restored as E2F1 also has a role in controlling the growth of HeLa cells, however we plan to test viability in cells where E2F1 is only partially depleted. Upon ATR inhibition in MCF7 the viability is fully rescued by siRNA of E2F1 (Figure R1E), suggesting that the DNA replication stress in *CCNF* K/O is mostly mediated by

E2F1 accumulation. We believe that expressing a mutant of E2F1 lacking the ‘*degron*’ and test the synthetic lethality using this mutant could further clarify this point.

8. In Figure 3E, E2F1 levels are identical between control and Cyclin F KO cells. Then in Figure 4A, they are hugely different (several fold). This is a major discrepancy.

The blot in Figure 3E is overexposed and differences in E2F1 are not readily appreciable. We will repeat the experiment and/or select a lower exposure of E2F1. Please note in all other figures (Figure 4A, 4B, 4F, EV3B, EV3G, EV3H, EV4A) where E2F1 signal is not overexposed, E2F1 levels are ‘*hugely*’ different. Thus, there is no major discrepancy.

9. The authors are far from having met the standard to show that E2F1 is a Cyclin F substrate. Missing experiments are many, and include, demonstration of E2F1 degradation upon Cyclin F overexpression, an assay for E2F1 ubiquitination, mapping of a degron on E2F1.

We present an assay for protein ubiquitylation which we conducted that shows that E2F1 is ubiquitylated by cyclin F (Figure R1A and B). We will optimise the *in vitro* assay during the revision process. We have mapped the cyclin F degron on E2F1 at an RxL in position 90-93.

10. When does Cyclin F control E2F1 in the cell cycle?

We have data indicating that cyclin F controls E2F1 in mitosis (Figure EV4D and EV4E).

11. The Chk1 inhibitor experiment in Figure 4F is not convincing, since the Cyclin F depleted cells have much more DNA damage, which is presumably driving E2F1 degradation. Moreover, it is not at all evident that the kinetics of E2F1 loss is greater, since the levels start higher.

It is possible to observe that *CCNF* K/O cells don’t have increased levels of γ -H2AX, pRPA S4/8, pKap1 S824 and pRPA S33 at time 0 (untreated cells) in figure 2A, 3E, 4F, EV3G. The comet assay presented in Figures 2L,2M,2N does not show increased DNA damage in *CCNF* K/O. *CCNF* K/O cells have consistently high levels of E2F1, which is in accordance to the main point of the manuscript. We now show that E2F1 reduction is not abrupt due to the concentration of Chk1 inhibitors used in the experiment in Figure 4F. When we use an higher concentration of Chk1i, E2F1 is reduced more efficiently. Importantly, we show that E2F1 is degraded after Chk1 inhibitor as the reduction of E2F1 triggered by Chk1i is controlled by the proteasome and neddylation (Figure R2A). The levels of E2F1 in *CCNF* K/O remains high in the presence of the same concentration of Chk1 inhibitors (Figure R2B).

12. The suggestion that phosphorylation of E2F1 controls degradation is not sufficiently supported.

We have identified an RxL motif that represents the ‘*degron*’ of cyclin F we will focus our experiments on this mutant rather than on identifying the phosphorylation of E2F1 promoting the interaction (Figure R2C).

Referee #2 (Report for Author)

We would like to thank the overall positive comments on our work:

‘the basic SL discovery between Cyclin F(CCNF) and CHK1i is exciting.’ ‘.....the manuscript provides a significant contribution to the field.....’

---Major points:

*Figure 2 is very extensive and rather complicated to read. It would benefit from a more visibly clear way of presenting the data (2B, C, D), while some of the data might probably be better kept as supplementary (i.e. figure 2E, F, G) for clarity of the figure. Figure 2H+I are relatively more key experiments (than figure 2E, F, G) but need a better description and evaluation in the text. Also in general, the text description of this part does not make the data easier to read and understand.

We apologise for this. The main point here is that *CCNF* K/O cells go in S phase where DNA damage is accumulating. We will amend the figure as suggested and plot the graph in an easy to visualise format.

*The authors are likely right regarding induction of replication catastrophe, however, they do not show this properly (for example by displaying RPA vs γ H2AX plots in FACS profiles to show the exhaustion of RPA).

We will plot the γ -H2AX on RPA as requested to show exhaustion of RPA.

*The presentation of the MS data and their further use in figure 3 is incomplete. It appears that a list (table) of CCNF interaction partners is missing, at least for the G/S factors suggested from figure 3A. Readers are left with unclear justification for choosing these factors for rescue experiments. In the absence of such table, it seemingly appears that (in Figure 3B) authors attempt phenotype rescue with knockdown of a very biased pathway-selected set of factors and not more of the known targets of CCNF. RRM2, SLBP and B-Myb might well contribute to the phenotype but they were not included (and should be).

Please note the selection of candidates from the Mass Spectrometry is biased as we could not conduct this experiment with over 100 genes. The experiments in the manuscript show that E2F1 is one if not the main mediator of replication catastrophe in *CCNF* K/O cells. Unfortunately, since RRM2 is also an E2F1 target it will be quite complex to dissect the two and it will be the scope of a different manuscript from the laboratory. SLBP and B-myb have not been directly associated to rescue of DNA replication stress induced by ATR and Chk1 inhibitors, however their role will be tested in a revised version of the manuscript.

*Figure 3E: siE2F1 also reduces CCNF levels. This indicates that siE2F1 might have considerable indirect effects on the cell cycle that dampen the phenotype following CHK1i. This notion should be tested by carefully analyzing the cell cycle and replication catastrophe +/-siE2F1 +/-CCNF +/- Chk1i.

To exclude that this is only due to cell cycle arrest we reanalysed the experiment in Figure 3 to determine the levels of γ -H2AX only in S Phase cells based on DNA content. When analysing only S phase cells it is clear that siRNA of E2F1 suppresses the formation of γ -H2AX (Figure R1C).

In addition, it is worth considering that although siRNA of Treslin and MTBP has been reported to arrest cells in S phase (Boos et al., 2013), siRNA of Treslin and MTBP does not reduce DNA damage in *CCNF* K/O cells exposed to Chk1i.

Furthermore, challenging MCF7 cells with ATR inhibitors and combined siRNA of cyclin F and siRNA of E2F1 rescue the DNA damage defect (Figure EV3H and I) without impacting on cell cycle distribution (Figure R1D). The combined siRNA of cyclin F and siRNA of E2F1 in MCF7 fully rescued cell viability after ATR inhibitors (Figure R1E).

At the same time, we can't exclude the fact that the accumulation of E2F1 accelerates entry into S phase, which could be part of the mechanism of synthetic lethality. We believe that expressing a mutant of E2F1 lacking the 'degron' and test the synthetic lethality using this mutant could further clarify this point.

*Role of E2F1 and/or downstream targets of E2F1: This aspect is not very developed. siE2F1 rescue might partly be mediated through reestablishing Cyclin E regulation though these data are less convincing. There is only one blot showing a partial rescue without good loading control. This could be removed or investigated properly including assays to understand CDK deregulation (also with assays in the model systems +/-CCNF +/- CHK1i). If the data are removed, it would be important to show that E2F1 targets are deregulated, which could be done at mRNA level or with E2F1 reporter (Cyclin E protein levels do not appear upregulated in *CCNF* K/O cells).

We have now obtained data from 3'RNAseq of *CCNF* K/O cell lines. We will use these data to analyse how E2F target genes are regulated/deregulated. We will conduct DNA combing experiments and likely gain more insights in the control of replication origins in *CCNF* K/O.

*The degradation mechanism should be further explored, which domains are required for the interaction in *CCNF* and E2F1, respectively? Complementation experiments with MR and F-BOX mutant *CCNF* are crucial and should give insight into how *CCNF* directly regulates E2F1. This is important in the absence of non-degradable E2F1 variant. The data in EV4 (A, D, and E) are very exciting and also discussed in the text in great detail, as such this should be a real figure, potentially figure 5 with the complementation data mentioned earlier. E2F1 half-life in EV4A should be quantified.

We have now identified a 'degron' in E2F1 which prevents interaction with cyclin F we will use this mutant and repeat the functional experiments. Furthermore, more experiments using cyclin F mutants will be conducted.

*The involvement of phosphorylation is not well developed and could be omitted (EV4B and C).

We have identified an RxL motif that represents the degron of cyclin F (Figure R2C) we will focus our experiments on this mutant rather on identify the phosphorylation of E2F1 promoting the interaction.

*SKP2 is a published ubiquitin ligase for E2F1, does SKP2 contribute to the observed regulatory circuitry? This question is obvious due to the lack of clear mechanistic data on the *CCNF*-E2F1 degradation.

We will test Skp2 role after Chk1 inhibition. It is worth to mention that in a screen conducted in the laboratory siRNA of Skp2 did neither increase or decrease sensitivity to ATR and Chk1 inhibitors (*data not shown*). Thus, although Skp2 could regulate E2F1 its functional relevance after ATR and Chk1 inhibitors is limited.

---Additional points relating to the discussion:

*transcription and replication initiation site overlap aspects; explanations of this statement including references are missing for the role of E2F1 in the regulation of this process.

Please note the discussion will be modified on this aspect as requested. The manuscript describing this has only been recently published (Chen et al., 2019). E2F1 has not been directly involved but it is easy to envision it could be, given its central role in regulating cell cycle.

*In relation to clinical relevance, is CHK1i also displaying SL-interactions with RB-loss or Cyclin E amplification (in cells with wt-CCNF)?

It has already been shown that RB loss sensitises cells to Chk1i (Witkiewicz et al., 2018), this will be appropriately discussed in a revised version of the manuscript.

*Claim of CCNF inactivation in cancer appears without support in the form of data and references. This should be provided.

The data are deposited in cBioPortal at (<https://www.cbioportal.org>). The results come from multiple studies and, therefore I am confused on how to cite them all. I am looking forward to suggestions from reviewers and editors.

Referee #3 (Report for Author)

We would like to thank the reviewer for the overall positive comments, constructive criticism and encouraging remark: ‘The data on synthetic lethality and DNA damage in S phase is strong and may be important from a clinical perspective.’

Major points.

Only cancer cell lines are investigated. As in particular p53 has functional connections with both ATR-Chk1 and E2F1 it would be important to see if the main findings translate to non-transformed cells (p53 positive cancer cells are poor substitutes as pathways likely adapted).

We will test the mechanism of synthetic lethality in primary cells like RPE.

It is important to test whether comparable populations are assessed after E2F1 siRNA. Does E2F1 siRNA affect cell-cycle progression, in particular amount of cells entering S-phase?

To exclude that the rescue of cyclin F-Chk1 synthetic lethality is only due to cell cycle arrest we reanalysed the experiment in Figure 3 to determine the levels of γ -H2AX only in S Phase cells based on DNA content. When analysing only S phase cells it is clear that siRNA of E2F1 suppresses the formation of γ -H2AX (Figure R1C).

In addition, it is worth considering that although siRNA of Treslin and MTBP has been reported to arrest cells in S phase (Boos et al., 2013), siRNA of Treslin and MTBP does not reduce DNA damage in *CCNF* K/O cells exposed to Chk1i.

Furthermore, challenging MCF7 cells with ATR inhibitors and combined siRNA of cyclin F and siRNA of E2F1 rescue the DNA damage defect (Figure EV3H and I) without impacting on cell cycle distribution (Figure R1D). The combined siRNA of cyclin F and siRNA of E2F1 in MCF7 upon ATR inhibitors fully rescued cell viability after ATR inhibitors (Figure R1E).

At the same time, we can't exclude the fact that the accumulation of E2F1 accelerates entry into S phase, which could be part of the mechanism of synthetic lethality. We believe that expressing a mutant of E2F1 lacking the 'degron' and test the synthetic lethality using this mutant could further clarify this point.

The manuscript is sparsely referenced throughout. For example, the paragraph describing clinical relevance in the discussion contains many examples but no references. The text is also not quite put into context: That Cyclin F has been described to downregulate cell cycle transcription through B-Myb is not described (Klein, Nat Com 2015), and that E2F1 has been described to be targeted for degradation by other factors is not mentioned (Marti, NCB 1999 and others).

We will modify the text to include these references and restructure the discussion of the manuscript as requested.

I find figure 4G slightly confusing. The interactions leading to the arrows should be discussed/referenced. Why is CDK1 but not CDK2 included? Why is there no arrow between E2F1 and CDK activity? Especially as the authors argue that the function likely is partially through expression of Cyclins. Similarly, is the only interaction between CDK and E2F1 a negative one? At the same time, Cyclin F can directly target E2F1 regulated genes that may impact replication. Absence of phenotype after depletion of E2F1 is therefore not conclusive for that E2F1 is the sole target of Cyclin F relevant for replication stress after Chk1 inhibition. This should be discussed and I'd suggest modifying figure 4G.

The model is an extremely simplified view of the findings in the manuscript and it doesn't cover other findings. This model will be modified in a revised version of the manuscript where we will clarify the interaction between E2F1 and CDKs. We will also focus on the role of a non-degradable version of E2F1 in triggering DNA replication catastrophe. Furthermore, the expression of E2F1 target genes will be further examined in *CCNF* K/O and after Chk1i.

RRM2 is a downstream target of E2F1 and a substrate of cyclin F. We believe to dissect the role of E2F1 and RRM2 in the cellular responses to Chk1i and ATRi would require a separate study.

In discussion: "The mechanism of DNA replication catastrophe induced". First, maybe more accurate with "seen as similar to", rather than "seen as"? Second, to make this point it would be important to show replication stress in the absence of Chk1i. Although that may be selected against in

stable knock-out clones, it should become apparent in depletion experiments.

We will modify the discussion as suggested. We don't observe a drastic accumulation of DNA damage upon *CCNF* K/O and siRNA of cyclin F. However, siRNA of cyclin F could induce DNA replication stress and DNA damage below the levels of Wb detection. This will be discussed as suggested.

To really make the point that Cyclin F targets E2F1 for degradation, it would be good to investigate whether Cyclin F supports ubiquitination of E2F1.

We have now preliminary evidence that E2F1 is ubiquitylated by cyclin F (Figure R1A and B).

Minor points:

Fig EV4D, E. Are amount of mitotic cells similar? In particular did cells pass through mitosis in E?

This will be tested using FACS.

I suppose the authors tried to test the interaction of endogenous Cyc F and E2F1? This should at least be commented upon.

E2F1 is a transcription factor and E2F1 levels are quite low in cells. For this reason, we have not tried to test interaction with endogenous but we will do so in the revision process.

"Cyclin F regulates E2F1 solely in the G2/M phases". I'm not sure I follow what data that is based on? Cyclin F expression from other publications? Please specify/reference. It may be helpful (maybe also in relation to point on 4G above) describing this in relation to CDK1 being restricted until S/G2 transition by Chk1 (Saldivar, Science 2018; Lemmens, Mol Cell 2018).

We will add the suggested references and modify the text to clarify this point. Cyclin F protein levels peak in G2/M and the majority of substrates are degraded during this phase. RRM2 and E2F1 seems to require a CDK1 phosphorylation as well.

In figure 3E, is there no difference in E2F1 levels with or without Cyclin F? It would be helpful to include a lower exposure.

The blot in Figure 3E is overexposed and differences in E2F1 are not readily appreciable. We will repeat the experiment for consistency.

Fig 2 E,F. It would be helpful with examples of gating

The plot in Figure 2E and F will be gated.

I would find it helpful with some more data from the MS experiment as a figure.

We will add a table that highlights the candidates we have selected to test in the screen in Figure 3A.

Please include page and figure numbers to help reviewing.

Yes, apologies for this they will be included in the revised version of the manuscript.

Sincerely,

Vincenzo D'Angiolella

- BOOS, D., YEKEZARE, M. & DIFFLEY, J. F. 2013. Identification of a heteromeric complex that promotes DNA replication origin firing in human cells. *Science*, 340, 981-4.
- CHEN, Y. H., KEEGAN, S., KAHLI, M., TONZI, P., FENYO, D., HUANG, T. T. & SMITH, D. J. 2019. Transcription shapes DNA replication initiation and termination in human cells. *Nat Struct Mol Biol*, 26, 67-77.
- CHOUDHURY, R., BONACCI, T., WANG, X., TRUONG, A., ARCECI, A., ZHANG, Y., MILLS, C. A., KERNAN, J. L., LIU, P. & EMANUELE, M. J. 2017. The E3 Ubiquitin Ligase SCF(Cyclin F) Transmits AKT Signaling to the Cell-Cycle Machinery. *Cell Rep*, 20, 3212-3222.
- D'ANGIOLELLA, V., DONATO, V., FORRESTER, F. M., JEONG, Y. T., PELLACANI, C., KUDO, Y., SARAF, A., FLORENS, L., WASHBURN, M. P. & PAGANO, M. 2012. Cyclin F-mediated degradation of ribonucleotide reductase M2 controls genome integrity and DNA repair. *Cell*, 149, 1023-34.
- WILLIAMS, K. L., TOPP, S., YANG, S., SMITH, B., FIFITA, J. A., WARRAICH, S. T., ZHANG, K. Y., FARRAWELL, N., VANCE, C., HU, X., CHESI, A., LEBLOND, C. S., LEE, A., RAYNER, S. L., SUNDARAMOORTHY, V., DOBSON-STONE, C., MOLLOY, M. P., VAN BLITTERSWIJK, M., DICKSON, D. W., PETERSEN, R. C., GRAFF-RADFORD, N. R., BOEVE, B. F., MURRAY, M. E., POTTIER, C., DON, E., WINNICK, C., MCCANN, E. P., HOGAN, A., DAOUD, H., LEVERT, A., DION, P. A., MITSUI, J., ISHIURA, H., TAKAHASHI, Y., GOTO, J., KOST, J., GELLERA, C., GKAZI, A. S., MILLER, J., STOCKTON, J., BROOKS, W. S., BOUNDY, K., POLAK, M., MUNOZ-BLANCO, J. L., ESTEBAN-PEREZ, J., RABANO, A., HARDIMAN, O., MORRISON, K. E., TICOZZI, N., SILANI, V., DE BELLEROUCHE, J., GLASS, J. D., KWOK, J. B., GUILLEMIN, G. J., CHUNG, R. S., TSUJI, S., BROWN, R. H., JR., GARCIA-REDONDO, A., RADEMAKERS, R., LANDERS, J. E., GITLER, A. D., ROULEAU, G. A., COLE, N. J., YERBURY, J. J., ATKIN, J. D., SHAW, C. E., NICHOLSON, G. A. & BLAIR, I. P. 2016. CCNF mutations in amyotrophic lateral sclerosis and frontotemporal dementia. *Nat Commun*, 7, 11253.
- WITKIEWICZ, A. K., CHUNG, S., BROUGH, R., VAIL, P., FRANCO, J., LORD, C. J. & KNUDSEN, E. S. 2018. Targeting the Vulnerability of RB Tumor Suppressor Loss in Triple-Negative Breast Cancer. *Cell Rep*, 22, 1185-1199.

1st Editorial Decision

20th March 2019

Thank you for response letter to our referees' comments, which I now finally had a chance to study in detail. I appreciate your proposals for revising your manuscript and addressing the key concerns, and given also the included preliminary data already at hand, would in this light be happy to consider a revised version of the study further for EMBO Journal publication. Particularly important points will be the envisioned analyses of non-degradable E2F1 versions both in biochemical and functional assays, as well as major improvements of the presented ubiquitination assays to more decisively demonstrate E3-dependence of E2F1 ubiquitination and specifically with wild-type but not mutant Cyclin F. Furthermore, I feel that in response to referee 1 and 2's shared concern regarding the incomplete presentation of the interactomics data in Figure 3A, it will be important to properly include the whole mass spec data set as an "Expanded View Dataset", and not just excerpts/snapshots from it. Finally, regarding referencing of cancer data on CCNF inactivation, I suggest consulting www.cbioportal.org/faq#how-do-i-cite-the-cbioportal (where two general references for the portal are provided), and possibly contacting the database to inquire whether all individual datasets need to be cited as well. For how to cite datasets in EMBO J papers, please carefully look at our revised guidelines at: emboj.embopress.org/authorguide#referencesformat

1st Revision - authors' response

5th July 2019

In the revised version, we have addressed all of reviewers' comments with experiments. We believe that the experiments suggested by the reviewers have significantly improved the overall message of the manuscript and we hope this is now a good fit for *EMBO J*.

A point-to-point answer follows below:

Referee #1

We would like to thank the referee for reading carefully through our data and provide insightful comments which have significantly improved the manuscript and its message:

1. The authors show that Cyclin F KO HeLa cells are sensitive to Chk1 inhibition. However, these drug studies are not performed in the sort of concrete way that would be satisfying to someone doing pharmacology. If this is going to be a drug study, which is what the first half of this manuscript is, then they should be performing careful dose response curves over a large dose range and analyzing cell proliferation/numbers/etc. They approach this in Figure 1D, although these data are usually shown on a log scale. If the drug experiment was supportive of a finding in a study focusing on a mechanistic aspect of cell or molecular biology, this request would be unnecessary. But in this case, the first half of the manuscripts aims to showcase the sensitivity of Cyclin F KO cells to these drugs, and therefore, it should be done appropriately.

We now present the data in Figure 1D on double log scale as requested. Our study is focused on the identification of a novel synthetic lethality and a novel substrate of cyclin F promoting the observed synthetic lethality. We believe that further experiments on the pharmacological aspects of our discovery would be outside the scope of this manuscript.

2. The authors fail to report the full data set from the drug screen, and thus, one cannot assess whether the Chk1 inhibitor that scored is an outlier among other inhibitors to the same target. The full data set would need to be reported to appreciate the importance of the screen.

We have now included a Table 1, which summarises the top hits identified in our screen and the source data associated to the screen (Source Data 1).

3. As stated above, it is not clear why the authors are screening for vulnerabilities to Cyclin F loss. And why are they doing so in HeLa cells?

Cyclin F is a crucial regulator of cell cycle progression and genome stability.

Furthermore, mutations in cyclin F have been found in neurodegenerative diseases (Williams et al., 2016). Identifying vulnerabilities could help basic cell biologists dissect cyclin F functions in various biological settings revealing novel pathways where cyclin F mediated ubiquitylation is crucial.

The manuscript is not intended to provide clinical significance but rather provide new insights into the role of cyclin F and Chk1. The discovery that cyclin F targets E2F1 and is synthetic lethal with Chk1 is a fundamental biological discovery which could have implications in the clinical settings for the reasons outlined below.

Cancers with **diploid** truncating mutations in cyclin F have been reported and studies are deposited in cbiportal (<https://www.cbiportal.org>):

In these cancers, it is possible to state that cyclin F is lost. The overall somatic mutation frequency across cancers is 1.1 %, it remains to be assessed whether mutations in cyclin F impact its function. In addition to mutations, cyclin F mRNA and protein levels as well as activity could vary. Please note it was reported that cyclin F activity is controlled by AKT phosphorylation (Choudhury et al., 2017). AKT is a known cancer driver. Finally, amplifications of cyclin F have been reported in more than 3 % of breast cancers (<https://www.cbiportal.org>) (Gao et al., 2013, Cerami et al., 2012).

I have the opinion that targeting sporadic events (like cyclin F mutations) could elicit a durable response as the emergence of these mutations in cancer is already a process of selection. It remains to be assessed whether targeting tumors lacking cyclin F in the clinic will make a difference, but this is well beyond the scope of this manuscript.

4. Figure 2K is an essential piece of data, showing that drug sensitivity is lost when Cyclin F is re-introduced. In its current form it is not convincing. The authors should have shown a blot for Cyclin F, demonstrating that it was restored to near endogenous levels. Moreover, they need to demonstrate that this eliminates the drug sensitivities observed in Figure 1, and not just alters the degree of DNA damage.

We have tried to reintroduce cyclin F in these cell lines with different retroviral systems and, usually the levels of cyclin F are lower than endogenous as reported before (Mavrommati et al., 2018). For this reason, we have transiently expressed GFP-cyclin F and measured γ -H2AX in GFP positive cells with low cyclin F expression (Figure 3A and 3B). The levels of cyclin F are higher than endogenous cyclin F, however, the same phenotype of drug sensitivity is obtained in U-2-OS (Figure 1 G and H), RPE-1 (Figure EV1C) and MCF7 (Figure EV3I) upon siRNA of cyclin F, demonstrating that this is not an off-target effect of the sgRNA and not a clonal effect of the *CCNF* K/O line. Furthermore, we have now reintroduced E2F1 Wild -Type (WT) and a stable mutant of E2F1 (Δ RxL) in cell lines and observed that a non-degradable version of E2F1 promotes more DNA damage (Figure 6E) and more cell death (Figure 6F) upon Chk1 inhibitors.

5. They indicate that IP-MS was done on Cyclin F, and that this enriched for proteins involved in cell cycle. These data would need to be shown to interpret this suggestion/implication. (Related to Figure 3A).

We agree with the reviewer that the gene enrichment set was confusing. For this reason we now provide a Table 2, which summarises the high confidence hits we have identified in our LC/MS experiment in the absence/presence of MLN4924, a general inhibitor of Cullin Ring ubiquitin Ligases (Soucy et al., 2009). From this list, we have selected interacting partners that were previously involved in DNA replication stress and sensitivity to Chk1i. In the mass spectrometry we identified E2F3, however loss of RB was previously shown to sensitise cells to Chk1i through E2F1 accumulation (Witkiewicz et al., 2018). Therefore, in our small screen we selected E2F1, E2F3, CDC6, CDT1, Geminin, Treslin and MTBP (Figure 4A and 4B).

6. A simple explanation for the rescue of DNA damage in E2F1 depleted cells would be that they are now arrested in G1. This is not addressed.

To exclude that this is only due to cell cycle arrest we reanalysed the experiment in Figure 3 to determine the levels of γ -H2AX only in S Phase cells based on DNA content. When analysing only S phase cells it is clear that siRNA of E2F1 suppresses the formation of γ -H2AX (Figure 4C).

In addition, it is worth considering that although siRNA of Treslin and MTBP has been reported to arrest cells in S phase (Boos et al., 2013), siRNA of Treslin and MTBP does not reduce DNA damage in *CCNF* K/O cells exposed to Chk1i.

Furthermore, challenging MCF7 cells with ATR inhibitors and combined siRNA of cyclin F and siRNA of E2F1 rescue the DNA damage defect (Figure EV3G and EV3H) without impacting on cell cycle distribution. The combined siRNA of cyclin F and siRNA of E2F1 in MCF7 treated with ATR inhibitors fully rescued cell viability (Figure EV3I).

Finally, we have now reproduced the increased DNA damage (Figure 6E) and cell death (Figure 6F) by expressing E2F1 WT and a non-ubiquitylatable (Figure 6B)/non-degradable version (Figure 6C) of E2F1 (Δ RxL). The last experiment demonstrates that increased stability of E2F1 promotes DNA damage and cell death upon Chk1 inhibition.

7. Importantly, the authors don't show that depleting E2F1 rescues the sensitivity to Chk1 inhibitors, but instead, that it alters the degree of DNA damage. This is key given the title of the paper.

The viability of HeLa cells upon siRNA of E2F1 and Chk1 inhibition is not fully restored as E2F1 also has a role in controlling the growth of HeLa cells. Upon ATR inhibition in MCF7 the viability is fully rescued by siRNA of E2F1 (Figure EV3I), suggesting that the DNA replication stress in *CCNF* K/O is mostly mediated by E2F1 accumulation. We now observe a significant increase of sensitivity to Chk1i (Figure 6F) upon expression of a non-degradable form of E2F1 (Δ RxL), demonstrating that E2F1 accumulation (due to lack of ubiquitylation by cyclin F) promotes cell death after Chk1 inhibitors.

8. In Figure 3E, E2F1 levels are identical between control and Cyclin F KO cells. Then in Figure 4A, they are hugely different (several fold). This is a major discrepancy. The blot in Figure 4E was overexposed and differences in E2F1 were not readily appreciable. We have now repeated the experiments and results are consistent with Figure 5A, 5B, 5F, EV3B, EV3G, EV4B, EV4D, EV4E.

9. The authors are far from having met the standard to show that E2F1 is a Cyclin F substrate. Missing experiments are many, and include, demonstration of E2F1 degradation upon Cyclin F overexpression, an assay for E2F1 ubiquitination, mapping of a degron on E2F1.

We now show that cyclin F WT but not cyclin F Δ F (lacking catalytic activity) and cyclin F M309A (lacking interaction with E2F1) is able to ubiquitylate E2F1 (Figure 5E). The half-life of E2F1 is increased in *CCNF* K/O compared to HeLa parental cells (Figure EV4D). Furthermore, the ubiquitylation of E2F1 is promoted by Chk1 inhibition and reduced upon siRNA of cyclin F (Figure 5I). Finally, a mutant of E2F1 lacking the motif of interaction with cyclin F (Δ RxL) is not ubiquitylated (Figure 6B) and has increased half-life (Figure 6C and quantified in D).

10. When does Cyclin F control E2F1 in the cell cycle?

Cyclin F is promoting E2F1 degradation in G2/M when cyclin F levels peak (Figure EV4B). A mutant of E2F1 lacking the interaction domain with cyclin F, E2F1 (Δ RxL) is not degraded in G2/M (Figure EV5B).

11. The Chk1 inhibitor experiment in Figure 4F is not convincing, since the Cyclin F depleted cells have much more DNA damage, which is presumably driving E2F1 degradation. Moreover, it is not at all evident that the kinetics of E2F1 loss is greater, since the levels start higher.

It is possible to observe that *CCNF* K/O cells don't have increased levels of γ -H2AX, pRPA S4/8, pKap1 S824 and pRPA S33 at time 0 (untreated cells) in Figure 2A, 4E, EV3G. The comet assay presented in Figures 3C 3D, and 3E does not show increased DNA damage in *CCNF* K/O.

We now show that E2F1 is degraded after Chk1 inhibitor as the reduction of E2F1 triggered by Chk1i is controlled by the proteasome and neddylation (Figure EV4E). The levels of E2F1 in *CCNF* K/O remains high in the presence of the same concentration of Chk1 inhibitors (Figure EV4E). Since E2F1 is expressed from a retroviral promoter and levels of E2F1 WT are reduced upon Chk1i in Figure 6E, we can state that the degradation of E2F1 is induced by Chk1i. The degradation of E2F1 is not occurring in cells expressing E2F1 Δ RxL.

12. The suggestion that phosphorylation of E2F1 controls degradation is not sufficiently supported.

We have identified an RxL motif that represents the 'degron' of cyclin F and have focused experiments on this mutant.

Referee #2 (Report for Author)

We would like to thank the overall positive comments on our work:

'the basic SL discovery between Cyclin F(CCNF) and CHK1 is exciting.' '.....the manuscript provides a significant contribution to the field.....'

---Major points:

*Figure 2 is very extensive and rather complicated to read. It would benefit from a more visibly clear way of presenting the data (2B, C, D), while some of the data might probably be better kept as supplementary (i.e. figure 2E, F, G) for clarity of the figure. Figure 2H+I are relatively more key experiments (than figure 2E, F, G) but need a better description and evaluation in the text. Also in general, the text description of this part does not make the data easier to read and understand.

We apologise for this. We have modified the graphs and described this section as suggested by the reviewer.

*The authors are likely right regarding induction of replication catastrophe, however, they do not show this properly (for example by displaying RPA vs γ H2AX plots in FACS profiles to show the exhaustion of RPA).

We have now plotted the γ -H2AX on RPA as requested (Figure 2H). The upturned shape of the plot of the nuclear ratio of RPA2 to γ -H2AX following Chk1 inhibition treatment resembles closely the described induction of DNA replication catastrophe (Toledo et al., 2013).

*The presentation of the MS data and their further use in figure 3 is incomplete. It appears that a list (table) of CCNF interaction partners is missing, at least for the G/S factors suggested from figure 3A. Readers are left with unclear justification for choosing these factors for rescue experiments. In the absence of such table, it seemingly appears that (in Figure 3B) authors attempt phenotype rescue with knockdown of a very biased pathway-selected set of factors and not more of the known targets of CCNF. RRM2, SLBP and B-Myb might well contribute to the phenotype but they were not included (and should be).

Please note the selection of candidates from the Mass Spectrometry is biased as we could not conduct this experiment with all the hits identified. The experiments in the manuscript pinpoint that E2F1 is one of the main mediators of DNA replication catastrophe in *CCNF* K/O cells. Indeed, the expression of a non-degradable version of E2F1 Δ RxL promotes more DNA damage and cell death after Chk1 inhibition (Figure 6E and 6F). Unfortunately, since RRM2 is also an E2F1 target it will be quite complex to dissect the two and it will be the scope of a different manuscript from the laboratory. SLBP and B-myb have not been directly associated to rescue of DNA replication stress induced by ATR and Chk1 inhibitors and their role will be interesting to test, however, at the moment we believe this experiment could subtract from the main focus of the manuscript.

*Figure 3E: siE2F1 also reduces CCNF levels. This indicates that siE2F1 might have

considerable indirect effects on the cell cycle that dampen the phenotype following CHK1i. This notion should be tested by carefully analyzing the cell cycle and replication catastrophe +/-siE2F1 +/-CCNF +/- Chk1i.

To exclude that this is only due to cell cycle arrest we reanalysed the experiment in Figure 3 to determine the levels of γ -H2AX only in S Phase cells based on DNA content. When analysing only S phase cells it is clear that siRNA of E2F1 suppresses the formation of γ -H2AX (Figure 4C).

In addition, it is worth considering that although siRNA of Treslin and MTBP has been reported to arrest cells in S phase (Boos et al., 2013), siRNA of Treslin and MTBP does not reduce DNA damage in *CCNF* K/O cells exposed to Chk1i.

Furthermore, challenging MCF7 cells with ATR inhibitors and combined siRNA of cyclin F and siRNA of E2F1 rescue the DNA damage defect (Figure EV3G and EV3H) without impacting on cell cycle distribution. The combined siRNA of cyclin F and siRNA of E2F1 in MCF7 treated with ATR inhibitors fully rescued cell viability (Figure EV3I).

Finally, we have now reproduced the increased DNA damage (Figure 6E) and cell death (Figure 6F) by expressing E2F1 WT and a non-ubiquitylatable (Figure 6B)/non-degradable version (Figure 6C) of E2F1 (Δ RxL). The last experiment demonstrates that increased stability of E2F1 promotes DNA damage and cell death upon Chk1 inhibition.

*Role of E2F1 and/or downstream targets of E2F1: This aspect is not very developed. siE2F1 rescue might partly be mediated through reestablishing Cyclin E regulation though these data are less convincing. There is only one blot showing a partial rescue without good loading control. This could be removed or investigated properly including assays to understand CDK deregulation (also with assays in the model systems +/-CCNF +/- CHK1i). If the data are removed, it would be important to show that E2F1 targets are deregulated, which could be done at mRNA level or with E2F1 reporter (Cyclin E protein levels do not appear upregulated in *CCNF* K/O cells). We have now obtained data from 3'RNAseq of *CCNF* K/O cell lines. We have uploaded the full dataset which shows a clear enrichment of E2Fs and E2F1 target genes (Figure 5C and Figure EV4A; Source Data 2).

*The degradation mechanism should be further explored, which domains are required for the interaction in *CCNF* and E2F1, respectively? Complementation experiments with MR and F-BOX mutant *CCNF* are crucial and should give insight into how *CCNF* directly regulates E2F1. This is important in the absence of non-degradable E2F1 variant. The data in EV4 (A, D, and E) are very exciting and also discussed in the text in great detail, as such this should be a real figure, potentially figure 5 with the complementation data mentioned earlier. E2F1 half-life in EV4A should be quantified. We agree that this was a weak point of the manuscript which we have now significantly strengthen. We now show that cyclin F WT but not cyclin F Δ F (lacking catalytic activity) and cyclin F M309A (lacking interaction with E2F1) is able to ubiquitylate E2F1 (Figure 5E). The half-life of E2F1 is increased in *CCNF* K/O compared to HeLa parental cells (Figure EV4D). Furthermore, the ubiquitylation of E2F1 is promoted by Chk1 inhibition and reduced upon siRNA of cyclin F (Figure 5I). Finally, a mutant of E2F1 lacking the motif of interaction with cyclin F (Δ RxL) is not ubiquitylated (Figure 6B) and has increased half-life (Figure 6C and quantified in D).

*The involvement of phosphorylation is not well developed and could be omitted (EV4B and C).

We have identified an RxL motif that represents the degron of cyclin F (Figure 6) we have focused our experiments on this mutant of E2F1.

*SKP2 is a published ubiquitin ligase for E2F1, does SKP2 contribute to the observed regulatory circuitry? This question is obvious due to the lack of clear mechanistic data on the CCNF-E2F1 degradation.

Our experiments show that cyclin F is the main regulator of E2F1 during cell cycle progression (Figure EV4B and EV5C) and after Chk1 inhibition (Figure 5F, 5G, 5H, 5I, EV5A and 6E). Although other ligases could be operating E2F1 ubiquitylation, we believe investigating their role would be the scope of a different manuscript.

---Additional points relating to the discussion:

*transcription and replication initiation site overlap aspects; explanations of this statement including references are missing for the role of E2F1 in the regulation of this process.

Please note the discussion has been modified on this aspect as requested. The manuscript describing this has only been recently published (Chen et al., 2019). E2F1 has not been directly involved but it is easy to envision it could be, given its central role in regulating cell cycle.

*In relation to clinical relevance, is CHK1i also displaying SL-interactions with RB-loss or Cyclin E amplification (in cells with wt-CCNF)?

It has already been shown that RB loss sensitises cells to Chk1i (Witkiewicz et al., 2018), this is discussed in a revised version of the manuscript.

*Claim of CCNF inactivation in cancer appears without support in the form of data and references. This should be provided.

The data are deposited in cBioPortal at (<https://www.cbioportal.org>), references to the database have been added as requested.

Referee #3 (Report for Author)

We would like to thank the reviewer for the overall positive comments, constructive criticism and encouraging remark: 'The data on synthetic lethality and DNA damage in S phase is strong and may be important from a clinical perspective.'

Major points.

Only cancer cell lines are investigated. As in particular p53 has functional connections with both ATR-Chk1 and E2F1 it would be important to see if the main findings translate to non-transformed cells (p53 positive cancer cells are poor substitutes as pathways likely adapted).

We have tested the mechanism in RPE cells (non-transformed cell lines) and knockdown of cyclin F by siRNA also promotes more cell death upon Chk1 inhibition (Figure EV1C).

It is important to test whether comparable populations are assessed after E2F1 siRNA.

Does E2F1 siRNA affect cell-cycle progression, in particular amount of cells entering S-phase?

To exclude that the rescue effect of E2F1 siRNA was due to cell cycle arrest we reanalysed the experiment in Figure 3 to determine the levels of γ -H2AX only in S Phase cells based on DNA content. When analysing only S phase cells it is clear that siRNA of E2F1 suppresses the formation of γ -H2AX (Figure 4C). In addition, it is worth considering that although siRNA of Treslin and MTBP has been reported to arrest cells in S phase (Boos et al., 2013), siRNA of Treslin and MTBP does not reduce DNA damage in *CCNF* K/O cells exposed to Chk1i. Furthermore, challenging MCF7 cells with ATR inhibitors and combined siRNA of cyclin F and siRNA of E2F1 rescue the DNA damage defect (Figure EV3G and EV3H) without impacting on cell cycle distribution. The combined siRNA of cyclin F and siRNA of E2F1 in MCF7 treated with ATR inhibitors fully rescued cell viability (Figure EV3I). Finally, we have now reproduced the increased DNA damage (Figure 6E) and cell death (Figure 6F) by expressing E2F1 WT and a non-ubiquitylatable (Figure 6B)/non-degradable version (Figure 6C) of E2F1 (Δ RxL). The last experiment demonstrates that increased stability of E2F1 promotes DNA damage and cell death upon Chk1 inhibition.

The manuscript is sparsely referenced throughout. For example, the paragraph describing clinical relevance in the discussion contains many examples but no references. The text is also not quite put into context: That Cyclin F has been described to downregulate cell cycle transcription through B-Myb is not described (Klein, Nat Com 2015), and that E2F1 has been described to be targeted for degradation by other factors is not mentioned (Marti, NCB 1999 and others).

We have now modified the text to include references as requested.

I find figure 4G slightly confusing. The interactions leading to the arrows should be discussed/referenced. Why is CDK1 but not CDK2 included? Why is there no arrow between E2F1 and CDK activity? Especially as the authors argue that the function likely is partially through expression of Cyclins. Similarly, is the only interaction between CDK and E2F1 a negative one? At the same time, Cyclin F can directly target E2F1 regulated genes that may impact replication. Absence of phenotype after depletion of E2F1 is therefore not conclusive for that E2F1 is the sole target of Cyclin F relevant for replication stress after Chk1 inhibition. This should be discussed and I'd suggest modifying figure 4G.

We have also provided a graphical abstract on the finding and omitted the role of CDKs which was not fully investigated. We also modified the discussion to clarify that other targets in addition to E2F1 could participate in the phenotype.

RRM2 is a downstream target of E2F1 and a substrate of cyclin F. We believe to dissect the role of E2F1 targets, like RRM2, in the cellular responses to Chk1i and ATRi would require a separate study. The mechanism of how oncogenes like cyclin E and E2F1 promotes DNA replication stress is the subject of intense debate in the field of DNA replication stress and is now discussed.

In discussion: "The mechanism of DNA replication catastrophe induced". First, maybe more accurate with "seen as similar to", rather than "seen as"? Second, to make this point it would be important to show replication stress in the absence of Chk1i. Although that may be selected against in stable knock-out clones, it should become

apparent in depletion experiments.

We have modified the discussion as suggested. We don't observe a drastic accumulation of DNA damage upon *CCNF* K/O and/or siRNA of cyclin F.

To really make the point that Cyclin F targets E2F1 for degradation, it would be good to investigate whether Cyclin F supports ubiquitination of E2F1.

We now show that cyclin F WT but not cyclin F Δ F (lacking catalytic activity) and cyclin F M309A (lacking interaction with E2F1) is able to ubiquitylate E2F1 (Figure 5E). The half-life of E2F1 is increased in *CCNF* K/O compared to HeLa parental cells (Figure EV4D). Furthermore, the ubiquitylation of E2F1 is promoted by Chk1 inhibition and reduced upon siRNA of cyclin F (Figure 5I). Finally, a mutant of E2F1 lacking the motif of interaction with cyclin F (Δ RxL) is not ubiquitylated (Figure 6B) and has increased half-life (Figure 6C and quantified in D).

Minor points:

Fig EV4D, E. Are amount of mitotic cells similar? In particular did cells pass through mitosis in E?

We have now clarified the cell cycle regulation of E2F1 in Figures EV4B and EV5B.

I suppose the authors tried to test the interaction of endogenous Cyc F and E2F1? This should at least be commented upon.

E2F1 is a transcription factor and E2F1 levels are quite low in cells. For this reason, we have not tried to test interaction with endogenous. However, the data obtained using a non-degradable version of E2F1 (Δ RxL) at near physiological levels support strongly the model that the regulation of E2F1 is mediated by cyclin F interaction through the RxL motif (89-91).

"Cyclin F regulates E2F1 solely in the G2/M phases". I'm not sure I follow what data that is based on? Cyclin F expression from other publications? Please specify/reference. It may be helpful (maybe also in relation to point on 4G above) describing this in relation to CDK1 being restricted until S/G2 transition by Chk1 (Saldivar, Science 2018; Lemmens, Mol Cell 2018).

Cyclin F is promoting E2F1 degradation in G2/M when cyclin F levels peak (Figure EV4B). A mutant of E2F1 lacking the interaction domain with cyclin F, E2F1 (Δ RxL) is not degraded in G2/M (Figure EV5C). References have been added as requested.

In figure 3E, is there no difference in E2F1 levels with or without Cyclin F? It would be helpful to include a lower exposure.

The blot in the previous Figure 3E is overexposed and differences in E2F1 were not readily appreciable. We have repeated the experiment plotted in the updated Figure 4E

Fig 2 E,F. It would be helpful with examples of gating

We have now simplified Figure 2 and plotted gated cells in Figure 2H.

I would find it helpful with some more data from the MS experiment as a figure.
We have included a Table 2 which summarises the most significant interacting partners identified in cyclin F LC/MS.

Please include page and figure numbers to help reviewing.
Pages and figure numbers are now included in the revised version of the manuscript.

Sincerely,
Vincenzo D'Angiolella

- BOOS, D., YEKEZARE, M. & DIFFLEY, J. F. 2013. Identification of a heteromeric complex that promotes DNA replication origin firing in human cells. *Science*, 340, 981-4.
- CERAMI, E., GAO, J., DOGRUSOZ, U., GROSS, B. E., SUMER, S. O., AKSOY, B. A., JACOBSEN, A., BYRNE, C. J., HEUER, M. L., LARSSON, E., ANTIPIN, Y., REVA, B., GOLDBERG, A. P., SANDER, C. & SCHULTZ, N. 2012. The cBio cancer genomics portal: an open platform for exploring multidimensional cancer genomics data. *Cancer Discov*, 2, 401-4.
- CHEN, Y. H., KEEGAN, S., KAHLI, M., TONZI, P., FENYO, D., HUANG, T. T. & SMITH, D. J. 2019. Transcription shapes DNA replication initiation and termination in human cells. *Nat Struct Mol Biol*, 26, 67-77.
- CHOUDHURY, R., BONACCI, T., WANG, X., TRUONG, A., ARCECI, A., ZHANG, Y., MILLS, C. A., KERNAN, J. L., LIU, P. & EMANUELE, M. J. 2017. The E3 Ubiquitin Ligase SCF(Cyclin F) Transmits AKT Signaling to the Cell-Cycle Machinery. *Cell Rep*, 20, 3212-3222.
- GAO, J., AKSOY, B. A., DOGRUSOZ, U., DRESDNER, G., GROSS, B., SUMER, S. O., SUN, Y., JACOBSEN, A., SINHA, R., LARSSON, E., CERAMI, E., SANDER, C. & SCHULTZ, N. 2013. Integrative analysis of complex cancer genomics and clinical profiles using the cBioPortal. *Sci Signal*, 6, p11.
- MAVROMMATI, I., FAEDDA, R., GALASSO, G., LI, J., BURDOVA, K., FISCHER, R., KESSLER, B. M., CARRERO, Z. I., GUARDAVACCARO, D., PAGANO, M. & D'ANGIOLELLA, V. 2018. beta-TrCP- and Casein Kinase II-Mediated Degradation of Cyclin F Controls Timely Mitotic Progression. *Cell Rep*, 24, 3404-3412.
- SOUCY, T. A., SMITH, P. G., MILHOLLEN, M. A., BERGER, A. J., GAVIN, J. M., ADHIKARI, S., BROWNELL, J. E., BURKE, K. E., CARDIN, D. P., CRITCHLEY, S., CULLIS, C. A., DOUCETTE, A., GARNSEY, J. J., GAULIN, J. L., GERSHMAN, R. E., LUBLINSKY, A. R., MCDONALD, A., MIZUTANI, H., NARAYANAN, U., OLHAVA, E. J., PELUSO, S., REZAEI, M., SINTCHAK, M. D., TALREJA, T., THOMAS, M. P., TRAORE, T., VYSKOCIL, S., WEATHERHEAD, G. S., YU, J., ZHANG, J., DICK, L. R., CLAIBORNE, C. F., ROLFE, M., BOLEN, J. B. & LANGSTON, S. P. 2009. An inhibitor of NEDD8-activating enzyme as a new approach to treat cancer. *Nature*, 458, 732-6.
- TOLEDO, L. I., ALTMAYER, M., RASK, M. B., LUKAS, C., LARSEN, D. H., POVLSSEN, L. K., BEKKER-JENSEN, S., MAILAND, N., BARTEK, J. & LUKAS, J. 2013. ATR

prohibits replication catastrophe by preventing global exhaustion of RPA. *Cell*, 155, 1088-103.

- WILLIAMS, K. L., TOPP, S., YANG, S., SMITH, B., FIFITA, J. A., WARRAICH, S. T., ZHANG, K. Y., FARRAWELL, N., VANCE, C., HU, X., CHESI, A., LEBLOND, C. S., LEE, A., RAYNER, S. L., SUNDARAMOORTHY, V., DOBSON-STONE, C., MOLLOY, M. P., VAN BLITTERSWIJK, M., DICKSON, D. W., PETERSEN, R. C., GRAFF-RADFORD, N. R., BOEVE, B. F., MURRAY, M. E., POTTIER, C., DON, E., WINNICK, C., MCCANN, E. P., HOGAN, A., DAOUD, H., LEVERT, A., DION, P. A., MITSUI, J., ISHIURA, H., TAKAHASHI, Y., GOTO, J., KOST, J., GELLERA, C., GKAZI, A. S., MILLER, J., STOCKTON, J., BROOKS, W. S., BOUNDY, K., POLAK, M., MUNOZ-BLANCO, J. L., ESTEBAN-PEREZ, J., RABANO, A., HARDIMAN, O., MORRISON, K. E., TICOZZI, N., SILANI, V., DE BELLEROCHE, J., GLASS, J. D., KWOK, J. B., GUILLEMIN, G. J., CHUNG, R. S., TSUJI, S., BROWN, R. H., JR., GARCIA-REDONDO, A., RADEMAKERS, R., LANDERS, J. E., GITLER, A. D., ROULEAU, G. A., COLE, N. J., YERBURY, J. J., ATKIN, J. D., SHAW, C. E., NICHOLSON, G. A. & BLAIR, I. P. 2016. CcNF mutations in amyotrophic lateral sclerosis and frontotemporal dementia. *Nat Commun*, 7, 11253.
- WITKIEWICZ, A. K., CHUNG, S., BROUGH, R., VAIL, P., FRANCO, J., LORD, C. J. & KNUDSEN, E. S. 2018. Targeting the Vulnerability of RB Tumor Suppressor Loss in Triple-Negative Breast Cancer. *Cell Rep*, 22, 1185-1199.

Accepted

29th July 2019

Thank you for submitting your final revised manuscript for our consideration. I am pleased to inform you that we have now accepted it for publication in The EMBO Journal.

REFERE REPORTS

Referee #2:

This manuscript version is very considerably improved. It is now convincingly addressing the mechanism of CCNF regulation of E2F1. Further, the authors have addressed all my major points, hence, I have no additional points of concern.

Corresponding Author Name: Vincenzo D'Angiolella

Journal Submitted to: The EMBO Journal

Manuscript Number: EMBOJ-2018-101443R